# Estimating ground-level CO concentrations across China based on national monitoring network and MOPITT: Potentially overlooked CO hotspots in the Tibetan Plateau

Dongren Liu [a], Baofeng Di [a,b], Yuzhou Luo [c], Xunfei Deng [d], Hanyue Zhang [a], Fumo Yang [a,e], Michael L. Grieneisen [c], Yu Zhan [a,e,f,g] *

[a] Department of Environmental Science and Engineering, Sichuan University, Chengdu 610065, China

[b] Institute for Disaster Management and Reconstruction, Sichuan University, Chengdu 610200, China

[c] Department of Land, Air, and Water Resources, University of California, Davis, CA 95616, United States

[d] Institute of Digital Agriculture, Zhejiang Academy of Agricultural Sciences, Hangzhou 310021, China

[e] National Engineering Research Center for Flue Gas Desulfurization, Chengdu 610065, China

[f] Sino-German Centre for Water and Health Research, Sichuan University, Chengdu 610065, China

[g] Medical Big Data Center, Sichuan University, Chengdu 610041, China

*Corresponding to*: Yu Zhan (yzhan@scu.edu.cn)

**Abstract.** Given its relatively long lifetime in the troposphere, carbon monoxide (CO) is commonly employed as a tracer for characterizing airborne pollutant distributions. The present study aims to estimate the spatiotemporal distributions of ground-level CO concentrations across China during 2013-2016. We refined the random-forest-spatiotemporal-kriging (RF-STK) model to simulate the daily CO concentrations on a 0.1° grid based on the extensive CO monitoring data and the Measurements of Pollution in the Troposphere CO retrievals (MOPITT-CO). The RF-STK model alleviated the negative effects of sampling bias and variance heterogeneity on the model training, with cross-validation $R^2$ of 0.51 and 0.71 for predicting the daily and multiyear average CO concentrations, respectively. The national population-weighted average CO concentrations were predicted to be $0.99 \pm 0.30$ mg m$^{-3}$ ($\mu \pm \sigma$) and showed decreasing trends over all regions of China at a rate of $-0.021 \pm 0.004$ mg m$^{-3}$ per year. The CO pollution was more severe in North China ($1.19 \pm 0.30$ mg m$^{-3}$), and the predicted patterns were generally consistent with MOPITT-CO. The hotspots in the Central Tibetan Plateau where the CO concentrations were underestimated by MOPITT-CO were apparent in the RF-STK predictions. This comprehensive dataset of ground-level CO concentrations is valuable for air quality management in China.

## 1 Introduction

Ground-level carbon monoxide (CO) is a worldwide atmospheric pollutant posing risks to human health and the environment (White et al., 1990; Reeves et al., 2002). While CO is formed naturally from the oxidation of methane and non-methane volatile organic compounds, anthropogenic emissions from incomplete combustion of fossil fuels and biofuels contribute approximately 42% of the total atmospheric CO (Holloway et al., 2000; Pommier et al., 2013). In spite of the slow decrease in CO concentrations in recent years based on satellite retrievals (Xia et al., 2016; Zheng et al., 2018), China is still one of the countries with the most severe CO pollution in the world, and the combustion of fossil fuels is the dominant source of anthropogenic CO emissions (Wang et al., 2004; Duncan et al., 2007a). Due to its relatively long lifetime in the troposphere (i.e., one to two months), CO is commonly employed as a tracer for characterizing pollutant transport in the atmosphere (Goldan et al., 2000; Pommier et al., 2010). It is therefore essential to obtain the spatiotemporal distribution of CO for air quality management. The national air pollution monitoring network in mainland China has been regularly observing ground-level CO concentrations since 2013 (MEPC, 2017) by the non-dispersive infrared absorption method and the gas filter correlation infrared absorption method (CNEMC, 2013), but these site-based measurements are inadequate to represent the spatially continuous distributions of CO (Xu et al., 2014).

Chemical Transport Models (CTMs) have been employed to estimate ground-level CO concentrations (Arellano and Hess, 2006; Hu et al., 2016). On the basis of meteorological conditions generated by climate models, CTMs simulate reactions, transport, and deposition of chemicals in the atmosphere, which generally require high computational cost and a large amount of data inputs such as emission inventories. The predictive performance of CTMs tends to be affected by uncertainties in the simulation algorithms and the emission inventories (Li et al., 2010; Hu et al., 2017a). A CTM comparison study found that the difference in transport simulation resulted in considerable discrepancies between inter-model CO predictions (Arellano and Hess, 2006; Duncan et al., 2007b). It has been reported that a certain CTM underpredicted the monthly average CO concentrations in China by more than 60% (Hu et al., 2016). Although the emission inventories for China have been refined in recent years, high uncertainties still exist (Li et al., 2017). For instance, biomass combustion, residential biofuel consumption, and

transient fire events tend to be underreported, consequently leading to underestimation of CO emissions in the emission inventories (Wang et al., 2002; Streets et al., 2003). Despite underestimation by CTMs, the general patterns of CO concentrations are captured, and they can be used as the a priori for deriving posterior estimates based on satellite retrievals (Deeter et al., 2014).

Multiple satellite instruments have been operating to measure atmospheric CO for more than a decade, including the Measurements of Pollution in the Troposphere (MOPITT) (Deeter et al., 2003; Worden et al., 2013a; Jiang et al., 2015; Deeter et al., 2017), the Atmospheric Infrared Sounder (McMillan, 2005; Wang et al., 2018), the Scanning Imaging Absorption Spectrometer for Atmospheric Chartography (Kopacz et al., 2010; Ul-Haq et al., 2016), and the Infrared Atmospheric Sounding Interferometer (Fortems-Cheiney et al., 2009; Barret et al., 2016). 

Strong absorption lines of CO occur in the thermal infrared (4.7 µm) and solar infrared (2.3 µm) spectral regions. Among the abovementioned satellite instruments, MOPITT is one of few sensors that are capable of measuring ground-level CO based on the instantaneous multispectral retrievals (Streets et al., 2013; Deeter et al., 2014; Deeter et al., 2017). The a priori used in MOPITT is simulated by the Community Atmosphere Model with Chemistry (CAM-Chem), which is a CTM. The MOPITT product plays an important role in analyzing

spatiotemporal patterns of ground-level CO at large scales (Drummond et al., 2010; Worden et al., 2013b; Strode et al., 2016). Compared with site-based in-situ monitoring, MOPITT provides repeated measures with more extensive spatial coverages. Nevertheless, the sensitivity of MOPITT signals to ground-level CO is affected by the thermal contrast between the ground and atmosphere (Warner et al., 2007; Clerbaux et al., 2009). High uncertainties in CO estimations retrieved from MOPITT have been reported, and more efforts are required to

improve the data quality (Zhao et al., 2006; Li and Liu, 2011).

Machine learning models have been applied to predict spatiotemporal distributions of atmospheric pollutants, such as fine particulate matter ($PM_{2.5}$) and nitrogen dioxide ($NO_2$), based on satellite retrievals and ground measurements (Reid et al., 2015; Zhan et al., 2018). Complex structures are built to capture nonlinear and high-order interactions between the response and predictor variables. Machine learning models generally show superior

predictive performance in the presence of abundant training data (Hastie et al., 2009). In the comparisons of models predicting $PM_{2.5}$ concentrations, random forests and gradient boosting machine, which incorporated satellite retrieved aerosol optical depth (AOD), presented conspicuously good predictive performance (Reid et al., 2015). In addition, the random forest and spatiotemporal kriging (RF-STK) model was proposed to predict the daily ground-level nitrogen dioxide ($NO_2$) concentrations across China based on satellite retrieved $NO_2$ densities

(Zhan et al., 2018). To the authors' knowledge, machine learning models have never been employed to estimate nationwide ground-level CO concentrations across China based on satellite retrievals.

The present study aims to estimate the spatiotemporal distributions of ground-level CO concentrations across China during 2013-2016. We refined the RF-STK model to simulate the daily gridded CO concentrations (0.1° grid with 98341 cells) based on the publicly available datasets, including the ground-level CO monitoring data,

the MOPITT retrieved surface CO (MOPITT-CO), and the extensive geographic factors. The strategy of inversely weighting the training data by the local population densities was proposed to mitigate the effect of sampling bias towards populous areas for the monitoring network. The spatial resolution of 0.1° has been commonly used for estimating the nationwide distributions of air pollutants in China (Guo et al., 2016; Zhan et al., 2017; Hu et al.,

2017b). A machine learning model (i.e., the RF-STK model), for the first time, assimilated the MOPITT-CO with the extensive site-based in-situ CO observations in order to provide more solid information for air quality management. This data assimilation approach compensated the shortcomings of the satellite retrievals (i.e., high uncertainty) and the in-situ measurements (i.e., low spatial coverage) with each other's strengths (i.e., large spatial coverage and high accuracy, respectively), which is more effective and flexible than CTMs in utilizing these measurements. The results of this study are expected to be valuable for air quality management in China.

## 2 Materials and methods

### 2.1 Ground-level CO observations

Figure 1 shows the locations of the 1656 monitoring sites spread out over all of China, which monitored the ground-level CO concentrations (MEPC, 2017; EPAROC, 2017; EPDHK, 2017). Most of the sites were in the cities of the eastern China, leading to nonnegligible sampling biases. Hourly average CO concentrations (mg m$^{-3}$) were collected and cleaned by employing the "three sigma rule" that the values falling outside of $\mu \pm 3\sigma$ were considered outliers (Kazmier, 2003). Less than 0.01% of the hourly data (values higher than 20.2 mg m$^{-3}$) were excluded. The days with more than 12-hour observations were included as representative days, and approximately 1.67 million records of daily average CO concentrations were obtained for the subsequent analyses.

### 2.2 MOPITT-CO retrievals

The MOPITT operational gas correlation spectroscopy CO product (MOP02J.007), containing retrievals of surface CO mixing ratios, was obtained from the Atmospheric Science Data Center (ASDC, 2017). The MOPITT onboard the Terra satellite provides tropospheric CO density with global coverage every three days (Edwards et al., 2004). The CO surface mixing ratios from the Level-2 data product have a spatial resolution of 22 km at nadir. The Level-2 product has daytime and nighttime data fields, which are highly correlated ($r = 0.99$). This study chose the daytime data over the nighttime data, as the former exhibit higher correlations with the ground-level CO observations than the latter (Table 1). The overall bias of Version 7 is a few percent lower than Version 6 for the thermal infrared (TIR)-only, near infrared (NIR)-only, and TIR/NIR products at all levels (Deeter et al., 2014; Deeter et al., 2017). The TIR/NIR product, which features the maximum sensitivity to near-surface CO, was used throughout this study and hereafter referred to as MOPITT-CO. Through the temporal and spatial convolution with Gaussian kernels (Goodfellow et al., 2016), the MOPITT noise was filtered and the data gaps were filled, which were then resampled to the 0.1° grid. Briefly, the MOPITT-CO data for each grid cell were first processed with the temporal convolution, which were then processed with the spatial convolution day by day. Please refer to Section S.1 in the Supplementary Data for the mathematical equations.

According to the ideal gas law, we converted the unit of MOPITT-CO data from ppb (the unit presented in the MOPITT product) to mg m$^{-3}$ in order to be comparable with the CO observations from the monitoring network:

$$C = B \cdot P \cdot M / (R \cdot T) \tag{1}$$

where $C$ is the CO concentration in the unit of mg m$^{-3}$, $B$ is the CO concentration in the unit of ppb, $P$ is the atmospheric pressure (atm), $M$ is the molecular weight of CO (mg mol$^{-1}$), $R$ is the gas constant (0.082 L atm mol$^{-}$

[1] K[-1]), and $T$ is the atmospheric temperature (K). Note that the data of atmospheric pressure and temperature for the unit conversion are available in the MOPITT product.

In order to evaluate the dependence of the MOPITT surface retrievals on the a priori information, we also extracted the averaging kernels and the a priori information from the MOPITT product. For each averaging kernel (a matrix), the sum of the elements in the row associated with the surface layer of the CO profile (hereafter referred to as the row-sum value) measures the overall dependence of the MOPITT surface CO retrievals on the a priori information (Deeter, 2017). A small row-sum value indicates strong dependence of the MOPITT retrieval on the a priori information, i.e., low sensitivity of the actual MOPITT retrieval. Please refer to Section S.2 in the Supplementary Data for the explanation of the averaging kernels.

## 2.3 RF-STK model

The RF-STK model, consisting of a random forest (RF) submodel and a spatiotemporal Kriging (STK), was refined to predict the daily ground-level CO concentrations across China. The RF-STK model utilizes the strengths of both RF and STK, which showed the capability of predicting $NO_2$ concentrations (Zhan et al., 2018). The RF-STK prediction is the sum of the RF prediction and the STK interpolation:

$$Z(s,t) = R(s,t) + K(s,t) \qquad (2)$$

where $Z(s,t)$ denotes the predicted CO concentration at location $s$ and time $t$, $R(s,t)$ is the spatiotemporal trend estimated by the RF submodel, and the prediction residual of the RF submodel, i.e., $K(s,t)$, is then interpolated with the STK submodel.

The RF submodel is an ensemble of regression trees. The average predictions of all the trees are output as the RF prediction. In the process of growing each tree, a random training dataset is prepared through bootstrap resampling from the original training dataset, while a random subset of the predictors is chosen in order to reduce the inter-correlation among the trees. The best split is determined at each tree node, which contributes the largest decrease in the squared error. Please refer to Section S.3 in the Supplementary Data for the detailed description of the RF algorithm.

As the CO concentrations approximated a lognormal distribution, they were log transformed for variance stabilization (De'Ath and Fabricius, 2000). Leveraging variable selection was conducted based on the pre-experiments. The out-of-bag (OOB) errors (representing the RF prediction residuals) of the back-transformed RF predictions were filtered with the "three-sigma-rule" and subsequently interpolated with the STK submodel. Finally, the CO concentrations were predicted as the sums of the STK interpolations and back-transformed RF predictions. It is worth mentioning that the RF submodel was refined in the present study by inversely weighting each training sample with the surrounding population density to alleviate the effects of sampling bias towards populous areas for the monitoring network. The loss function ($L$) of the RF submodel is as follows:

$$L(y, f(x)) = \sum_{n=1}^{N} w_n [y_n - f(x_n)]^2 / \sum_{n=1}^{N} w_n \qquad (3)$$

where $w_n$ is the weight of observation $y_n$ ($N$ observations in total), and $f(x_n)$ is the model prediction.

## 2.4 Model input data

The predictors of environmental conditions for the RF-STK model covered the meteorological conditions, land uses, emission inventories, elevation, population densities, normalized difference vegetation index (NDVI), and road densities. The meteorological conditions included the atmospheric pressure, air temperature, precipitation, evaporation, relative humidity, insolation duration, wind speed, and planetary boundary layer height (PBLH). Land uses mainly recorded the areas of forests, grasslands, wetlands, artificial surfaces, and waterbodies. The emission inventories comprised emission distributions of ten major atmospheric chemical constituents, such as CO, organic carbon, and black carbon. The meteorological conditions, except for PBLH, were interpolated to the 0.1° grid by using co-kriging with elevation. The elevation, land uses, population densities, NDVI, PBLH, and emission inventories were resampled to the 0.1° grid by calculating area-weighted means, for which additional predictors were generated by applying spatial convolution with Gaussian kernels. The spatial convolution smoothed spatial transition and took into account neighboring effects (Goodfellow et al., 2016). Please refer to Section S.4 and Table S1 in the Supplementary Data for the detailed descriptions and data sources of the environmental conditions.

**2.5 Model evaluation**

The predictive performance and the predictor effects of the RF-STK model were investigated. We compared the predictive performance of the RF-STK models with/without the MOPITT data (either the a priori information or the MOPITT retrievals) by using two cross-validation strategies, including the site- and region-based cross-validation. With the 10-fold site-based cross-validation, all the monitoring sites were approximately evenly divided into ten groups. In each iteration, nine groups were used to develop a model, and the remaining group was used for validation. The training and prediction steps were repeated 10 times so that every ground-level CO observation had a paired prediction. While the site-based cross-validation is a commonly used strategy, it tends to overestimate the predictive performance given the fact that the monitoring sites tend to be clustered. Therefore, we also employed the region-based cross-validation strategy by following the concept of cluster-based cross-validation that was proposed to resolve the issue of clustered sites (Young et al., 2016). Different from the site-based cross-validation, the region-based cross-validation divided the training data by the geographic regions (e.g., North China and East China; Fig. 1) for the cross-validation. Various statistical metrics, such as the coefficient of determination ($R^2$), root mean square error (RMSE), and mean normalized error (MNE), were used to reflect the predictive performance. In addition, the measures of variable importance and partial dependence plots were employed to evaluate the predictor effects. The improvement in the split-criterion attributed to a predictor variable measured its relative importance in the model. A partial dependence plot illustrated the effect of a predictor on the CO concentrations after accounting for the average effects of all the other predictors (Friedman, 2001; Hastie et al., 2009).

**2.6 Spatiotemporal analyses**

Detailed spatiotemporal analyses were performed to investigate the correlation strength between the MOPITT data (including the a priori information and the MOPITT retrievals) and ground-level CO observations, as well as the distributions of the ground-level CO predictions. The whole nation was divided into seven conventional regions, including Central, East, North, Northeast, Northwest, South, and Southwest China (Fig. 1). For each region, the effectiveness of the MOPITT-CO was evaluated by estimating its correlation with the ground-level

CO observations at daily, seasonal, and annual scales. In addition, the seasonal/annual average concentrations maps were delineated based on the full-coverage CO predictions. The population-weighted averages of MOPITT-CO ($M_{PW}$) and ground-level CO predictions ($C_{PW}$) were summarized for the whole nation and by regions. The temporal trends of the national and regional $M_{PW}$ and $C_{PW}$ were evaluated by conducting linear regression on the time series of monthly averages that were deseasonalized by the loess smoothers (Cleveland, 1990). More detailed analyses were conducted for the North China Plain (NCP) and the Central Tibetan Plateau (CTP). While the air pollution in NCP has been well recognized, the air quality in CTP is usually considered to be pristine. Nevertheless, CTP was identified as a potentially overlooked CO hotspot in the present study.

**2.7 Computing environment**

The data processing and modeling were mainly performed using python and R (R Core Team, 2018). The *scikit-learn* python package was used to develop random forests (Pedregosa et al., 2012). The spatial operations, such as spatiotemporal kriging were conducted by using the R packages of *gstat* (Gräler et al., 2016), *rgdal* (Bivand et al., 2017), and *sp* (Pebesma and Bivand, 2005).

**3 Results and discussion**

**3.1 Descriptive statistics of CO measurements from monitoring network and MOPITT**

The ground-level CO observations from the monitoring network show that the average CO concentrations for China was $1.07 \pm 0.74$ mg m$^{-3}$ ($\mu \pm \sigma$) during 2013-2016. The ground-level CO observations approximated a lognormal distribution, with a median of 0.90 mg m$^{-3}$ and an interquartile range (IQR) of 0.69 mg m$^{-3}$. The hourly CO concentrations were the highest at 9am and the lowest at 4pm based on the average diurnal cycle (Fig. S1). High CO concentrations (daily average > 4.0 mg m$^{-3}$) were observed in 704 monitoring sites, with $7.6 \pm 0.8$ days per year (CREAS and CNEMC, 2012). The CO concentrations show a strong seasonality, ranging from $0.81 \pm 0.17$ mg m$^{-3}$ in summer to $1.39 \pm 0.38$ mg m$^{-3}$ in winter (Fig. S2). The national annual average of CO concentrations decreased by 6.9% from year 2013 to 2016 (Fig. 2). Note that the scale of monitoring network was not constant, and the number of monitoring sites grew from 743 to 1603 during these four years (MEPC, 2017; EPAROC, 2017; EPDHK, 2017). However, the monitoring stations were still sparse in the western China throughout the monitoring period, and most of the stations were located in the major cities of the eastern China (Fig. 1). The spatially imbalanced monitoring (i.e., sampling bias) therefore tends to introduce bias to the spatiotemporal statistics of CO concentrations (Boria et al., 2014). For instance, the national average concentrations would be overestimated if they were simply determined as the averages of all the monitoring data, as the CO concentrations were generally lower in remote areas.

The MOPITT-CO data, with an overall coverage rate of $3.5 \pm 0.5\%$, show that the surface CO level for China was $0.23 \pm 0.18$ mg m$^{-3}$ during 2013-2016 (Fig. S2). The MOPITT-CO values also approximated a lognormal distribution, with a median of 0.18 mg m$^{-3}$ and an IQR of 0.19 mg m$^{-3}$. The MOPITT-CO had the highest coverage in fall ($4.2 \pm 1.9\%$) and lowest in summer ($2.9 \pm 1.5\%$) (Table S2). Southwest China, especially the Sichuan Basin, had the lowest coverage rate (< 1%) in China (Fig. S3). In addition to the reflectance condition and the satellite orbit, the narrower swath width of MOPITT (640 km) compared to the Moderate Resolution Imaging

Spectroradiometer (MODIS) with a swath width of 2330 km was one of the main factors causing the sparse coverage. While MOPITT and MODIS are both onboard the Terra satellite, the measurement repeat cycle of MOPITT is approximately 3 days compared to 1-2 days of MODIS (Edwards et al., 2004). The sparse coverages of MOPITT-CO limit its utility for representing time-series of daily CO concentrations across China.

**3.2 MOPITT-CO evaluation against ground-level CO observations**

The spatiotemporal pattern of the MOPITT-CO was generally consistent with that of the ground-level CO observations in China, with $r = 0.43$ for the multiyear averages and $r = 0.37$ for the daily values during 2013-2016 (Table 1). The correlation between the a priori and the ground-level observations was weaker, with $r = 0.34$ for the multiyear averages and $r = 0.30$ for the daily values, suggesting that the MOPITT retrievals provided more information on the ground-level CO distributions than the a priori. The spatiotemporal distributions of the row-sum values of the averaging kernels demonstrate that the dependence of the MOPITT retrievals on the a priori varied widely (Fig. S4). Among the seven geographic regions of China, the average row-sum values during 2013-2016 were the highest in East China and the lowest in Northeast China. Seasonally, the national average row-sum values were the highest in fall and the lowest in summer/winter. The row-sum values were lower in CTP than NCP, suggesting a stronger dependence of the MOPITT retrievals on the a priori in CTP than NCP. The variations in the sensitivity of the MOPITT retrievals could result from various sources, such as the CO amounts and the diurnal temperature differences (Deeter et al., 2003; Deeter, 2007; Worden et al., 2013b).

The MOPITT-CO satisfactorily reflected the west-east spatial gradient and the seasonality (i.e., low in warm seasons and high in cold seasons) of ground-level CO concentrations (Figs. 3 and S5). Severe CO pollution in the eastern China resulted from the intensive anthropogenic emissions (Fig. S6). At both national and regional scales, the correlation coefficients between ground-level CO observations and MOPITT-CO were generally higher in winter than the other three seasons. The stronger correlation in winter was mainly attributed to the higher signal-to-noise ratios accompanied with the higher CO concentrations, reflecting that the MOPITT-CO was more sensitive in measuring high CO concentrations. In addition, the correlation strength of daily values exhibited considerable spatial heterogeneity, with $r$ ranging from 0.58 for South China to 0.17 for Southwest China (Table S3). As expected, it was difficult to capture the CO variations under highly complex geographic conditions in Southwest China, and the high uncertainty in the emission inventories undermined the representativeness of MOPITT-CO for that region. Especially for CTP, we found that the MOPITT-CO was almost completely insensitive to the variations of ground-level CO, with $r = -0.03$ in contrast to $r = 0.35$ for NCP (Table 1). The CO hotspots observed in the main cities of CTP (e.g., Naqu and Qamdo) were not recognized by MOPITT-CO, which even falsely showed the opposite seasonality of ground-level CO (Figs. 4 and S2).

The discrepancies between the MOPITT-CO and the ground-level CO observations could be mainly attributed to the low sensitivity of the satellite instrument to the ground-level CO variations and the high uncertainty associated with the a priori for deriving the MOPITT retrievals. The low sensitivity caused high uncertainties in the measured radiances (associated with the instrumental noises) and hence led to large measurement errors (ASDC, 2017). In addition, the accuracy of the a priori information was influenced by the data quality of the emission inventory and the sophistication of the CTM (i.e., the CAM-Chem model), which subsequently affected the accuracy of the posterior estimation (Dekker et al., 2017). The CO emission amounts for China were reported to be largely

underestimated (Streets et al., 2003; Wang et al., 2004), which might explain the fact that the MOPITT-CO was approximately half of the ground-level CO observations. Especially for CTP, the inadequate information about the CO emissions could be the main reason why MOPITT-CO largely underestimated the ground-level CO concentrations, whereas some relatively densely populated cities (such as Naqu and Qamdo; Fig. 1) had high CO concentrations (Chen et al., 2019). The population in Naqu and Qamdo are over one million, reflecting intensive anthropogenic activities (NBS, 2010). Biomass (e.g., yak dung) combustion, which is of low utilization efficiency, is widely used in CTP for energy, resulting in considerable CO emissions (Cai and Zhang, 2006; Wen and Tu, 2011; Xiao et al., 2015). Naqu is sandwiched between the Tanggula and the Nyainqen Tanglha Mountains (Fig. 1), which is unfavorable for CO dispersion and causes CO accumulation.

### 3.3 Predictive performance of the RF-STK model

On the basis of the site-based cross-validation results, the RF-STK model showed reasonable performance in predicting the daily ground-level CO concentrations, with $R^2$=0.51, RMSE=0.54 mg m$^{-3}$, and slope=0.64 (Fig. 5). Through the variable selection, a concise structure of the RF submodel was achieved, and the spurious prediction details (e.g., the sharp boundaries) were mitigated (Fig. S7). For instance, the RF submodel with all the predictors generated sharp boundaries circling the desert areas in Northwest China, which became blurred in the predictions made by the reduced RF submodel with the selected predictors (Fig. S8). Note that the coordinate variables (i.e., latitude and longitude) were not considered as candidate variables for the RF submodel, as artificial strips emerged in the prediction maps after including them as was illustrated in a previous study (Zhan et al., 2017). For the STK submodel, the predictions were further fine-tuned based on the spatiotemporal patterns of the RF submodel prediction residuals. As a result, the cross-validation slope increased from 0.55 to 0.64 (Table S4), suggesting an improvement in capturing the high and low concentrations.

Compared to the original RF-STK model proposed in the previous study (Zhan et al., 2018), this refined RF-STK model had two major modifications, including sample weighting and logarithm transformation of the response variable (i.e., ground-level CO observations in the present study). Inversely weighting the training samples by their surrounding population densities alleviated the effects of sampling bias towards populous areas for the monitoring network. As a result, the CO monitoring data from the sparsely populated areas (e.g., the Tibetan Plateau) gained higher weights in the model training process for compensating the scarcity of the training samples, leading to more realistic predictions for those areas. In addition, observations with higher variations would naturally gain higher weights during model training given the loss function of squared errors, for which it was suggested to transform the response variable to achieve homogeneity of variance (De'Ath and Fabricius, 2000). The ground-level CO observations were heavy-tailed distributed, and hence logarithm transformation was conducted prior to training the RF submodel. Compared with the original RF submodel, the refined RF submodel showed similar performance in the cross-validation but predicted more realistic spatial distributions of ground-level CO across China (Table S4 and Fig. S8). The spatial distributions predicted by the original RF submodel showed the prevalence of higher concentrations than those predicted by the refined RF submodel, resulting from overweighting of the training data from the areas with more serve CO pollution, e.g., NCP.

It is noteworthy that the RF-STK model with MOPITT-CO ($R^2$=0.51, RMSE=0.54 mg m$^{-3}$, and slope=0.64; Table 2) was superior to the model without MOPITT-CO ($R^2$=0.49, RMSE=0.58 mg m$^{-3}$, and slope=0.60; Table 2) and

the model with the a priori information ($R^2$=0.49, RMSE=0.57 mg m$^{-3}$, and slope=0.60; Table S4) based on the site-based cross-validation results (Tables 2 and S4). The performance difference became more apparent in the region-based cross-validation, where the model with MOPITT-CO ($R^2$=0.45, RMSE=0.61 mg m$^{-3}$, and slope=0.52) clearly outperformed the model without MOPITT ($R^2$=0.32, RMSE=0.69 mg m$^{-3}$, and slope=0.46). We therefore reasoned that the MOPITT-CO data were essential for the RF-STK model to achieve better predictive performance, especially for the areas without monitoring sites nearby.

As a machine learning approach, the RF-STK model exhibited stable performance across regions and seasons (Fig. S9), which was comparable or superior to the previous CTMs or statistical methods simulating ground-level CO concentrations (Table S5). As the simulation areas and episodes were considerably different among these studies, their predictive performance was not strictly comparable. A hybrid statistical model (partial least square and support vector machine) exhibited decent goodness-of-fit in simulating daily CO concentrations in Tehran, Iran, with fitting $R^2$=0.65 (Yeganeh et al., 2012). For the CTM study in Bahia, Brazil, the accuracy of the posterior estimation improved largely after incorporating the surface observations into the priori state (Hooghiemstra et al., 2012). In the absence of nationwide statistical modeling work, only CTM studies were found for modeling CO at large scale in China. A previous CTM work for China underestimated the ground-level CO concentrations by 67.2% on average (Hu et al., 2016), which might be due to the underestimation of CO emissions.

**3.4 Important predictors**

On the basis of the variable importance evaluation, MOPITT-CO was the most important predictor in the RF-STK model with relative importance of 9.4%, and the emission-related predictors together accounted for 30.0% of the total importance (Fig. 6). The partial dependence plots delineated the complicated relationships between the predictors and the ground-level CO concentrations, which could be difficult to be specified in parametric models (Fig. S10). While MOPITT-CO contained essential information for the RF-STK model to make accurate predictions, the high uncertainties pertaining to the MOPITT retrievals prevented the MOPITT-CO from playing a dominant role in the model, and the other predictors were also indispensable. Among the emission-related predictors, the spatial-convolution-processed emission of organic carbon was the most important predictor (importance: 8.5%), which reflected the spatiotemporal patterns of anthropogenic emissions from industrial and residential sectors (Fig. S11). Given the high intercorrelations among the predictors associated with anthropogenic emissions, only the most informative predictors were retained in the model after the variable selection (Figs. S6 and S12).

As the most important group of predictors, the meteorological conditions together accounted for 35.6% of the total importance (Fig. 6). The relative importance of temperature, evaporation, wind speed, atmospheric pressure, PBLH, relative humidity, and insolation duration ranged from 2.8 to 8.6%. In general, stagnant weather conditions occurred more frequently in winter, which was characterized by shallow mixed layers, less precipitation, and slow wind speed. These weather conditions caused accumulation of atmospheric pollutants discharged by local emissions or transported from outside, which aggravated local air pollution (Wang et al., 2014). Similar to other atmospheric pollutants, the CO concentrations were also sensitive to meteorological conditions (Xu et al., 2011). For instance, the apparently negative associations of the CO concentrations with the PBLH and the wind speed were delineated by the corresponding partial dependence plots (Fig. S10). Nevertheless, it should be noted that

the partial dependence plot illustrated the overall relationship and could be distorted by spatial and/or temporal confounders. For instance, the partial dependence plot for temperature, with a peak around 20°C, was contrary to the fact that the CO concentrations were the highest in winter. This "false" relationship was due to the phenomenon that most of the CO-polluted areas distributed in the warm zones of China, i.e., the spatial factor confounded the relationship between temperature and CO concentrations.

**3.5 Spatiotemporal distributions of ground-level CO predicted by the RF-STK model**

The RF-STK predictions showed similarly spatiotemporal patterns to MOPITT-CO while presented more fine-scale details (Figs. 3 and 7). The predictions of the RF-STK adequately assimilated the information of ground-level CO observations, with $r = 0.95$ for the daily concentrations (Table 1). The nationwide multiyear (i.e., 2013-2016) $C_{PW}$ were predicted to be $0.99 \pm 0.30$ mg m$^{-3}$, with the highest seasonal averages ($1.32 \pm 0.49$ mg m$^{-3}$) for winter and the lowest ($0.77 \pm 0.22$ mg m$^{-3}$) for summer (Fig. 2). The regional $C_{PW}$ were predicted to be the highest in North China and the lowest in South China, with the concentrations of $1.19 \pm 0.30$ and $0.77 \pm 0.18$ mg m$^{-3}$, respectively. It is worth noting that the RF-STK predictions showed the CO hotspots in CTP, where the ground-level CO concentrations were underestimated by MOPITT-CO (Fig. 8). The "abnormal" CO seasonality (i.e., low in winter and high in summer) for CTP characterized by the MOPITT-CO was corrected in the RF-STK predictions even though the data quality of ground-level CO observations for 2013 were in doubt (Fig. 4). The high CO concentrations in CTP might result from the low combustion efficiency of residential stoves and the large amount of biomass combustion for energy (Chen et al., 2015). For example, combustion of yak dung accounted for more than 50% of the energy consumption in Nagqu (Yang and Zheng, 2015).

During 2013-2016, the nationwide $C_{PW}$ decreased from $1.02 \pm 0.34$ to $0.95 \pm 0.30$ mg m$^{-3}$ at a rate of $-0.021 \pm 0.004$ mg m$^{-3}$ per year ($P<0.01$; Figs. 2 and 9). The relative decrease rate of 4.4% was similar to the 3.8% drop of coal consumption for China during 2013-2016, suggesting the potentially important contribution of decrease in coal consumption (partially due to improved energy conversion efficiency; Fig. S11) to the mitigation of CO pollution (CSY, 2018). Coal consumption accounted for approximately 70% of the total energy use in China. As the major energy consumers, the industrial and residential sectors contributed 41 and 39% of the total anthropogenic CO emissions, respectively (Fig. S13). More coal was consumed for residential heating in winter, causing higher CO emissions and more severe air pollution (Fig. S14). The relatively decreasing rate of CO was similar to that of NO$_2$ but much slower than the decreasing trend of PM$_{2.5}$ (Ma et al., 2016; Zhan et al., 2018). Spatially, the $C_{PW}$ significantly decreased for all regions ($P<0.05$) except for Southwest China ($P=0.16$). The decreasing trend was most prominent for North China where CO pollution was the most severe, with a decreasing rate of $-0.028 \pm 0.008$ mg m$^{-3}$ per year.

In comparison to the RF-STK predictions (which were very similar to ground-level CO observations given the good model fitness), the MOPITT-CO tended to underestimate the decreasing trends of ground-level CO concentrations (Fig. 9). The absolute decreasing rate of $M_{PW}$ for the whole China during 2013-2016 was approximately 60% lower than that of the RF-STK predictions (i.e., $C_{PW}$). The relative change rate of $M_{PW}$ was -1.99% compared to -2.25% of $C_{PW}$ per year. Spatially, the $M_{PW}$ showed no significant trends for East, Northeast, Northwest, South, and Southwest China ($P>0.05$). We found that the trend underestimation tended to be more severe for the regions with weaker averaging kernels, indicating higher dependence on the a priori information

(Figs. 9 and S5). In addition, the decreasing trend predicted by the RF-STK model with the a priori information (-2.06% per year) was slower than that predicted by the RF-STK model with the MOPITT-CO (-2.25% per year; Fig. S15). We thus deduced that the a priori information, which was the same across the years (Dekker et al., 2017), might greatly contribute to the trend underestimation by the MOPITT-CO.

The issue of bias drift for the MOPITT retrievals, which could result from long-term instrumental degradation (Deeter et al., 2017), should also be considered in the trend analyses. The bias drift for MOPITT-CO was found to be approximately -0.69% per year based on the flask measurements performed by the National Oceanic and Atmospheric Administration (Deeter et al., 2017). It is noteworthy that the extents of bias drift were of considerable spatial variation (Buchholz et al., 2017). For the present study, if the MOPITT-CO data were

"corrected" by the bias drift of -0.69%, the relative change rate of $M_{PW}$ would become lower (-1.31% per year), and the trend underestimation by the MOPITT would be more severe (Fig. S16). Accurate information on the temporal trends of CO is essential for air quality management, and more efforts are thus required to improve the data quality of CO measurements.

In order to advance the knowledge of ground-level CO distributions, the study period would be extended, and the

spatiotemporal resolution would be improved for future work. We chose the period of 2013-2016 due to the data availability in the beginning of 2018 when we started to conduct this study. While the air pollution in China was severer in earlier years (Krotkov et al., 2016), no large-scale monitoring data were available before 2013 for training the RF-STK model. Back-extrapolation such as a previous study (Gulliver et al., 2016) may be conducted based on MOPITT-CO since 2000, whereas the issue of bias drift is currently difficult to deal with. In addition,

measurements or model predictions with high spatial (e.g., 1 km) and temporal resolutions (e.g., 1 hour) are important to studies focusing on small regions, such as CTP in this study. In spite of its relative coarse resolution (22 km at nadir), the MOPITT product provided the best publicly available satellite-based measurements of surface CO for China during 2013-2016. Since July of 2018, the TROPOspheric Monitoring Instrument onboard the Sentinel-5P satellite has been providing the CO product at a higher resolution of 7 km × 3.5 km (Borsdorff et

al., 2018), which could replace MOPITT-CO in the RF-STK model in order to make predictions for years after 2018 at a higher resolution.

**4 Conclusions**

The spatiotemporal distributions of ground-level CO concentrations for China during 2013-2016 were derived by using the RF-STK model to assimilate the satellite and ground-based measurements. The RF-STK model showed

feasible performance in predicting the daily CO concentrations on the 0.1° grid. As most of the monitoring sites were in urban areas, we refined the RF-STK model through inversely weighting the training samples with the surrounding population densities. Given the fact of monitoring sites clustered in cities, it is critical to take into account the effects of sampling bias on modeling the spatiotemporal distributions of atmospheric pollutants. While the general patterns were well depicted by the MOPITT retrievals, the fine-scale distributions were sharpened and

corrected with the observations from the monitoring network. By using this data-fusion approach, we obtained the comprehensive dataset of ground-level CO concentrations for China.

On the basis of the spatiotemporal predictions, the population-weighted average of ground-level CO concentrations was $0.99 \pm 0.30$ mg m$^{-3}$ for China during 2013-2016, with a decreasing rate of $-0.021 \pm 0.004$ mg m$^{-3}$ per year. The CO concentrations were predicted to be the highest in North China ($1.19 \pm 0.30$ mg m$^{-3}$) and the lowest in South China ($0.77 \pm 0.18$ mg m$^{-3}$). The seasonal averages of the whole China ranged from $0.77 \pm 0.22$ in summer to $1.32 \pm 0.49$ mg m$^{-3}$ in winter, attributing to the seasonality of weather conditions and emission intensities as indicated by the variable importance of the RF-STK model. The present study provides important information for improving the accuracy of MOPITT retrievals, such as refining the a priori assigned to the CO hotspots in CTP constrained by the RF-STK predictions. The predicted results of ground-level CO distributions are valuable for air quality management and human exposure assessment in China.

*Code availability*. The code for random forest is available from scikit-learn (https://scikit-learn.org/stable/). The code for spatiotemporal kriging is available from the Comprehensive R Archive Network (https://cran.r-project.org/web/packages/gstat/index.html).

*Data availability*. The hourly CO concentration data are from the Ministry of Ecology and Environment of the People's Republic of China (http://datacenter.mep.gov.cn/). The MOPITT data are from the Atmospheric Science Data Center (https://eosweb.larc.nasa.gov/). The estimated ground-level CO concentrations are available upon request.

*Author contributions*. DL and YZ performed research and wrote manuscript. BD, YL, XD, and HZ analyzed data. FY and MLG provided extensive comments on manuscript.

*Competing interests*. The authors declare that they have no conflict of interests.

*Acknowledgements*. The authors would like to thank Dr. Daniel Jaffe at University of Washington-Bothell, Dr. Ya Tang and Guangming Shi at Sichuan University, and the anonymous referees for reviewing this manuscript. This research is supported by the National Natural Science Foundation of China (21607127, 41875162), the Fundamental Research Funds for the Central Universities (YJ201765), and the Sichuan "1000 Plan" Young Scholar Program.

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

**Table 1.** Correlations among the ground observations, MOPITT-CO, and the RF-STK predictions (Pearson correlation coefficients).

| Region/Dataset | Pair[a] | Daily | Monthly | Seasonal | Annual | Spatial[b] |
|---|---|---|---|---|---|---|
| Nation | O-M | 0.37 | 0.40 | 0.45 | 0.44 | 0.43 |
| | O-P | **0.95** | **0.97** | **0.97** | **0.97** | **0.98** |
| | P-M | 0.09 | 0.1 | 0.1 | 0.09 | 0.13 |
| Central Tibetan Plateau (CTP)[c] | O-M | -0.03 | -0.04 | 0.11 | -0.12 | -0.12 |
| | O-P | **0.91** | **0.92** | **0.93** | **0.96** | **1** |
| | P-M | -0.04 | -0.04 | -0.06 | -0.09 | -0.12 |
| North China Plain (NCP)[c] | O-M | 0.35 | 0.36 | 0.40 | 0.30 | 0.20 |
| | O-P | **0.95** | **0.97** | **0.98** | **0.97** | **0.98** |
| | P-M | 0.35 | 0.40 | 0.47 | 0.52 | 0.58 |
| X1_PRI[d] | O-M | 0.30 | 0.32 | 0.38 | 0.34 | 0.34 |
| X1_TS[d] | O-M | 0.39 | 0.47 | 0.49 | 0.44 | 0.42 |
| X2[d] | O-M | 0.37 | 0.39 | 0.45 | 0.42 | 0.40 |

[a] O: ground-level CO observations; M: MOPITT-CO; P: predictions made by the RF-STK model; the correlation coefficients higher than 0.90 are in **bold**.

[b] Multiyear averages during 2013-2016.

[c] Please refer to Fig. 1 for the locations of CTP and NCP.

[d] X1_PRI: nationwide a priori for MOPITT-CO; X1_TS: nationwide MOPITT-CO processed with the temporal and spatial convolution; X2: nationwide nighttime MOPITT-CO, and all the other MOPITT-CO data refer to daytime retrievals.

**Table 2.** Performance comparisons of the RF-STK models with/without MOPITT data in predicting daily ground-level CO concentrations across China during 2013-2016.

| Metric[a] | Site-based cross-validation[b] | | Region-based cross-validation[b] | |
|---|---|---|---|---|
| | With MOPITT | Without MOPITT | With MOPITT | Without MOPITT |
| $R^2$ | 0.51 | 0.49 | 0.45 | 0.32 |
| Slope | 0.64 | 0.60 | 0.52 | 0.46 |
| RMSE | 0.54 | 0.58 | 0.61 | 0.69 |
| RPE | 50.4% | 54.0% | 56.7% | 64.2% |
| MFB | -0.022 | -0.025 | -0.027 | 0.036 |
| MFE | 0.35 | 0.35 | 0.39 | 0.43 |
| MNB | 0.70 | 0.75 | 0.78 | 0.89 |
| MNE | 0.98 | 1.02 | 1.08 | 1.19 |

[a] $R^2$: coefficient of determination; RMSE: root mean square error (mg m$^{-3}$); RPE: relative prediction error; MFB: mean fractional bias; MFE: mean fractional error; MNB: mean normalized bias; MNE: mean normalized error. Bold: the best performance of each evaluation metric. Lower values are better for each metric except $R^2$ and slope.

[b] Site-based cross-validation: The training data are randomly divided into 10 groups stratified by the monitoring sites for the cross-validation. Region-based cross-validation: The training data are divided by the geographic regions (e.g., North China and East China; Fig. 1) for the cross-validation.

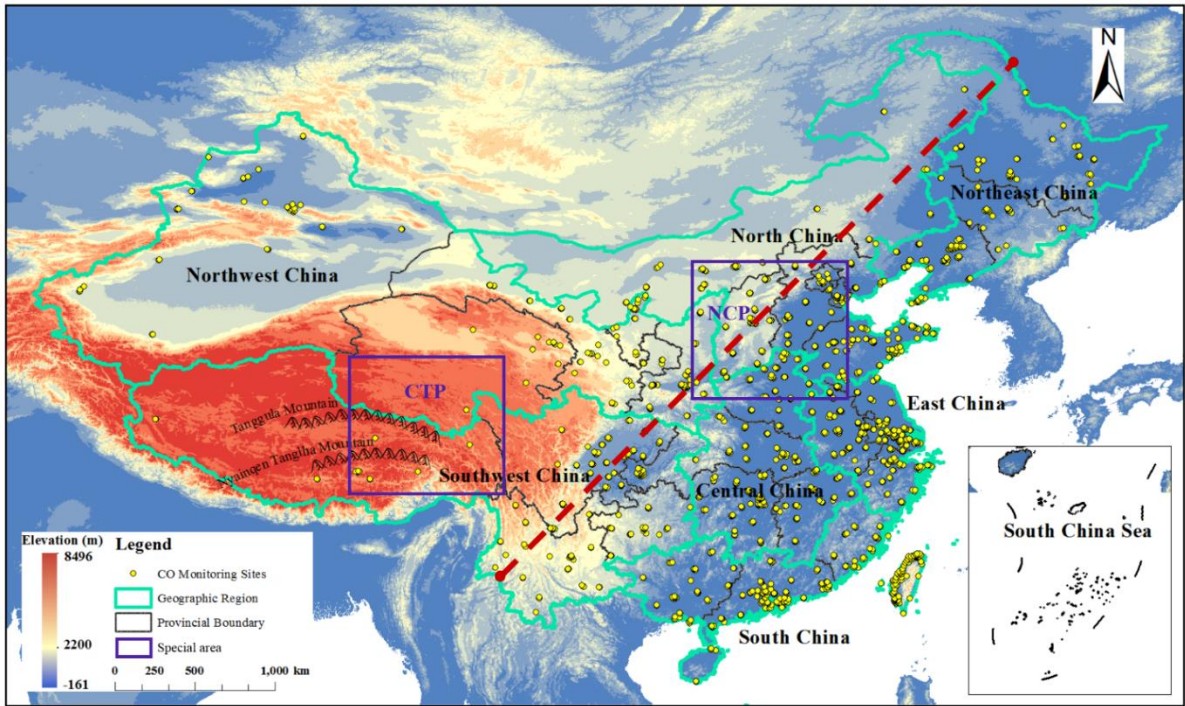

**Figure 1: Ground-level CO monitoring network for China in 2013-2016 with 1656 sites in total. The Central Tibetan Plateau (CTP) and the North China Plain (NCP) are labelled on the map. The red dashed line represents the Heihe-Tengchong Line, which is an imagined "geo-demographic demarcation line" reflecting the disparity in the population distribution. Around 95% of the population live to the east of the line, where 82% of the monitoring sites are located.**

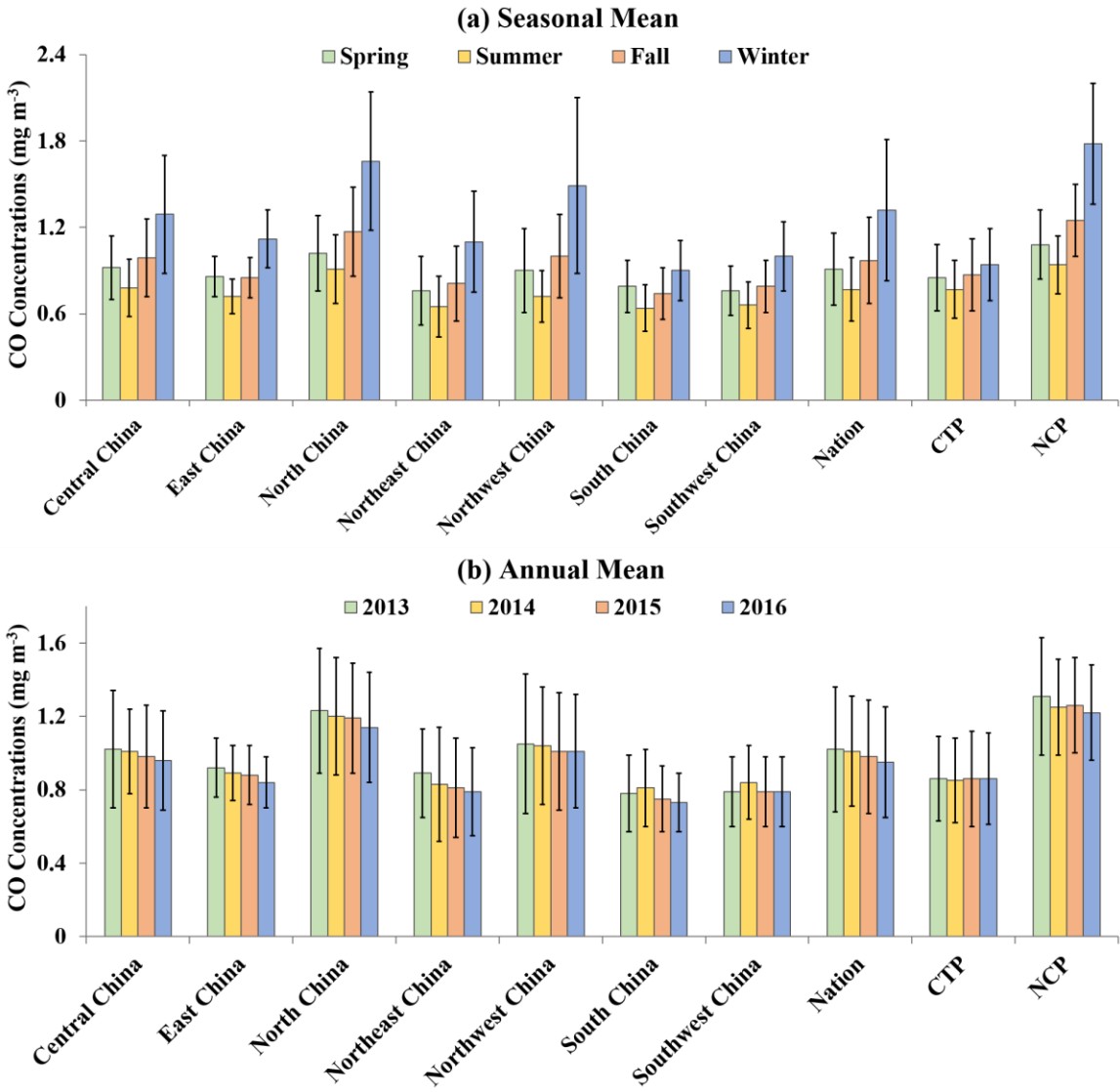

**Figure 2: (a) Seasonal and (b) annual means of the population-weighted average ground-level CO concentrations (mg m⁻³) during 2013-2016 for China predicted by the RF-STK model. The error bars (standard deviations) stand for the spatial variations.**

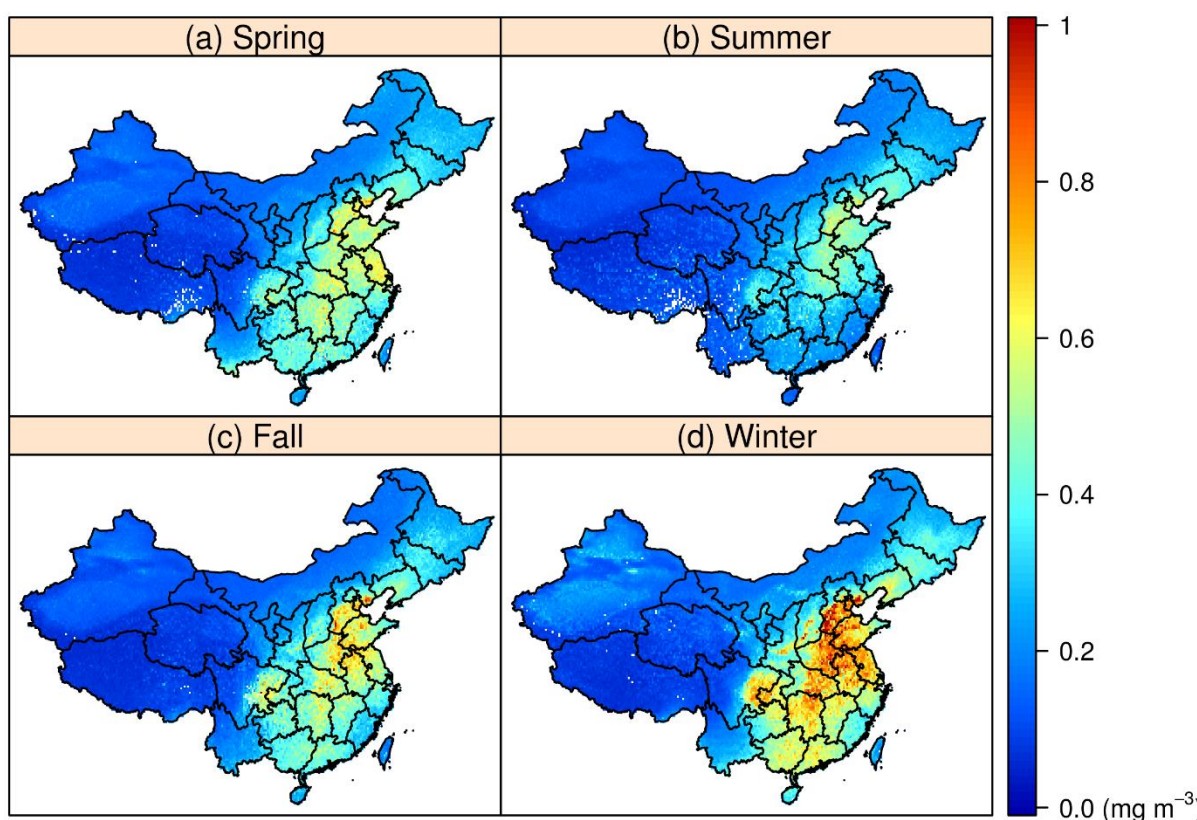

**Figure 3: Seasonal averages of the MOPITT retrieved surface CO concentrations (mg m$^{-3}$) in (a) spring, (b) summer, (c) fall, and (d) winter during 2013-2016 across China.**

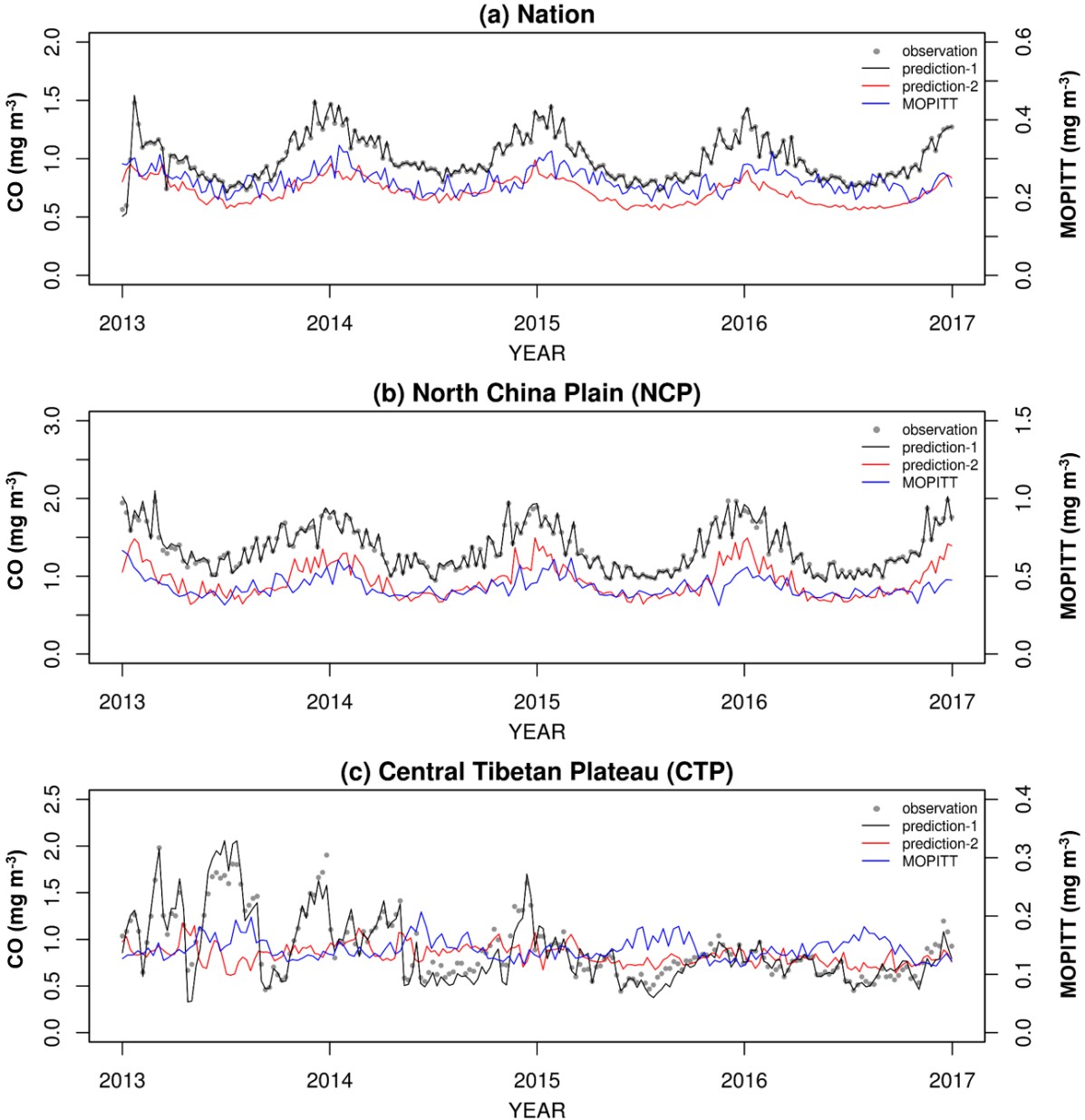

**Figure 4: Temporal variations of the average ground-level CO concentrations for (a) the whole nation, (b) the North China Plain (NCP), and (c) the Central Tibetan Plateau (CTP) during 2013-2016 based on the observations from the monitoring network (grey points), the RF-STK predictions (black and red solid lines), and the MOPITT retrievals (blue solid lines). The black lines show the RF-STK predictions for the grid cells with monitoring sites (prediction-1), and the red lines show the RF-STK predictions for all the grid cells (prediction-2). Weekly averages rather than daily concentrations are presented for clarity. Please refer to the right Y-axis for the MOPITT retrievals and the left Y-axis for all the other time series.**

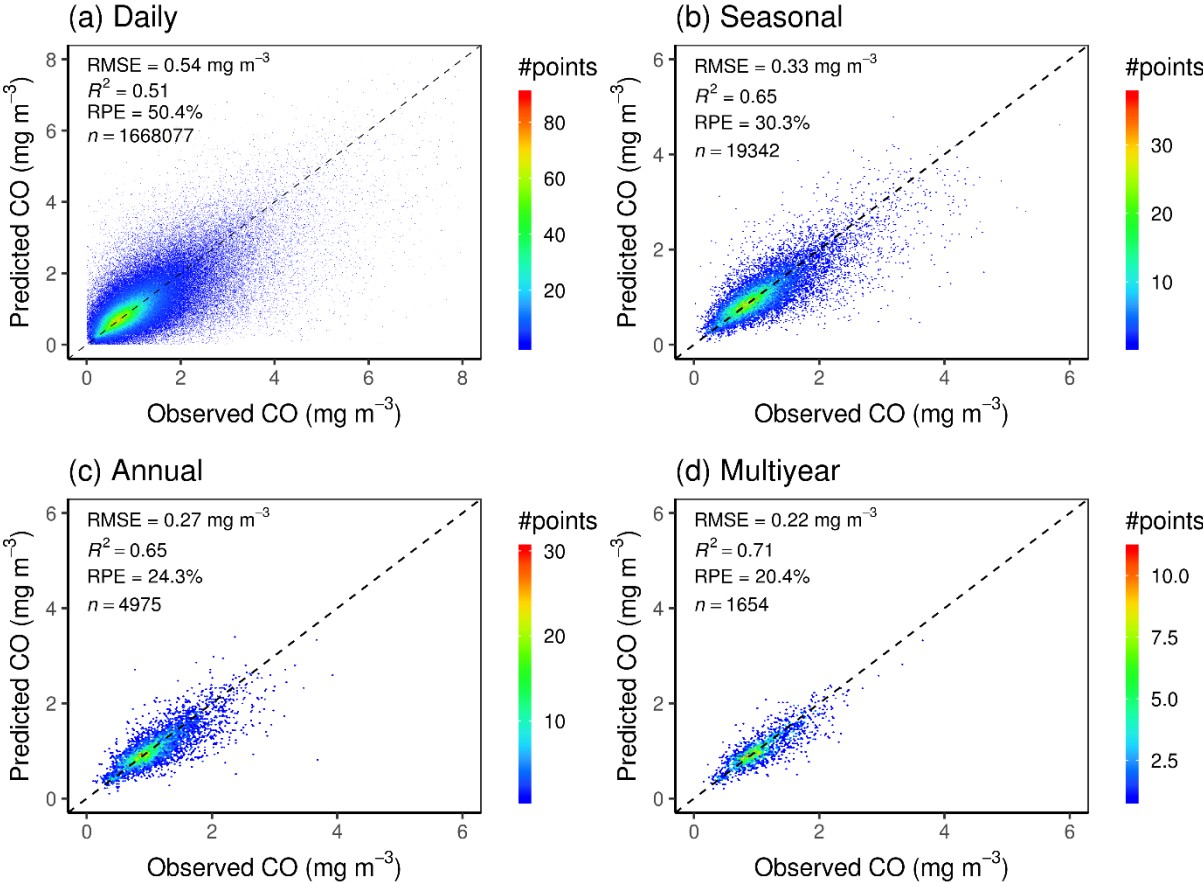

**Figure 5: Performance of the RF-STK model in predicting (a) daily, (b) seasonal, (c) annual, and (d) spatial (i.e., multiyear average) ground-level CO concentrations across China during 2013-2016. The dashed lines represent the 1:1 relationship.**

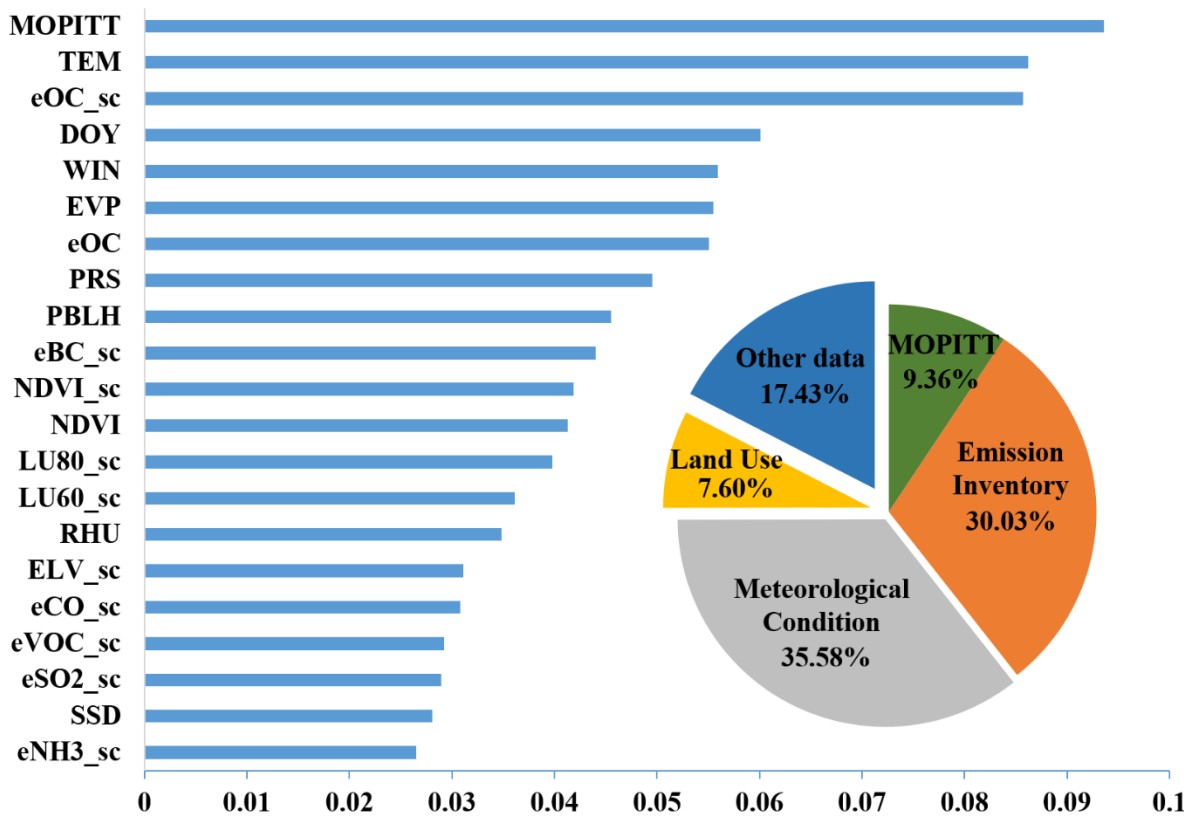

**Figure 6: Relative importance of the predictor variables in the RF-STK model. Please refer to Table S1 for the detailed descriptions of these variables.**

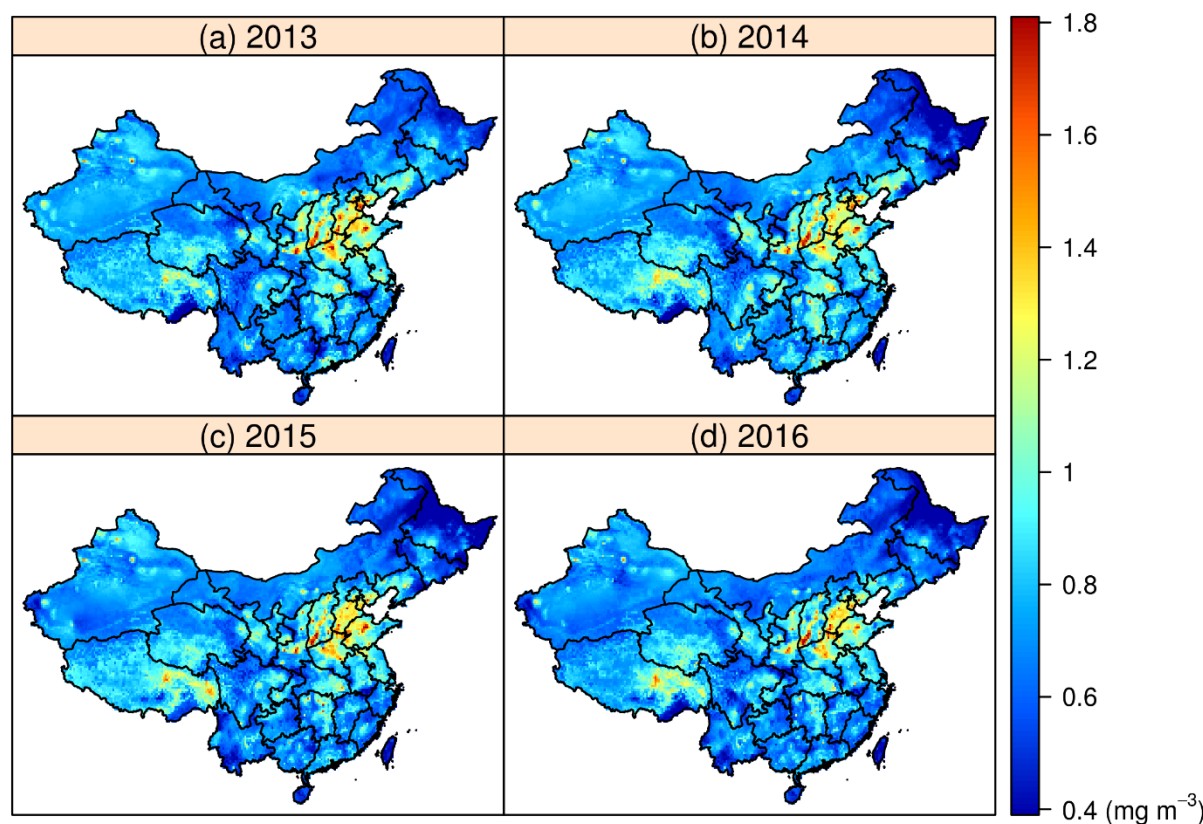

**Figure 7: Annual average ground-level CO concentrations predicted by the RF-STK model for (a) 2013, (b) 2014, (c) 2015, and (d) 2016 across China.**

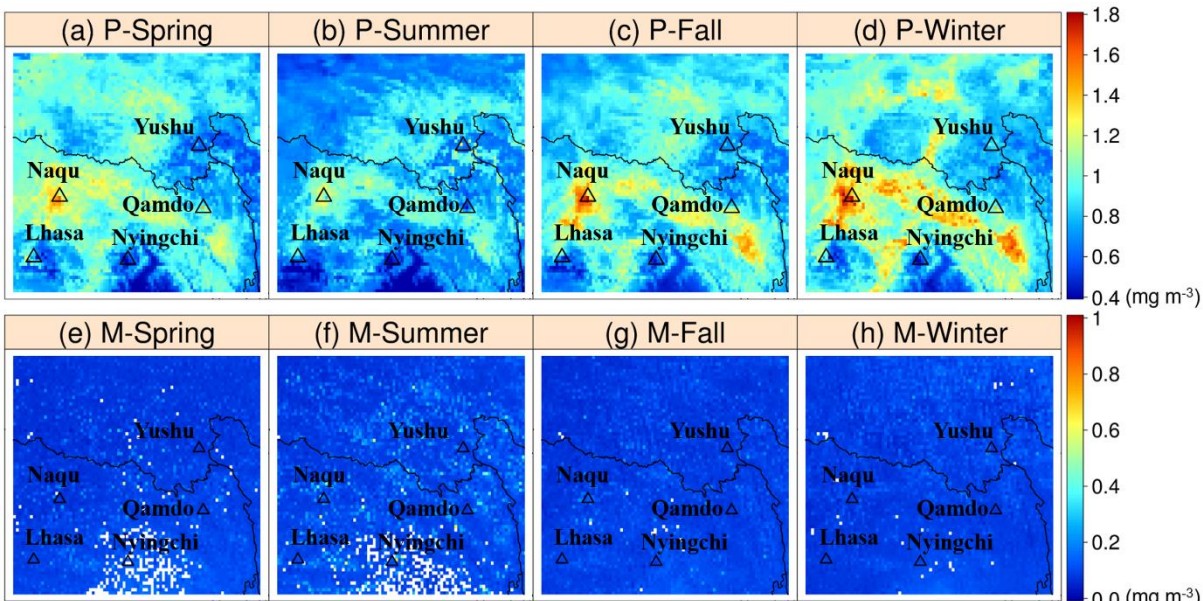

**Figure 8: Seasonal average ground-level CO concentrations (mg m⁻³) during 2013-2016 in the Central Tibetan Plateau based on (a-d) the RF-STK predictions (P) and (e-h) the MOPITT retrievals (M). Main cities within this area (e.g., Lhasa, Naqu, and Qamdo) are annotated with triangles.**

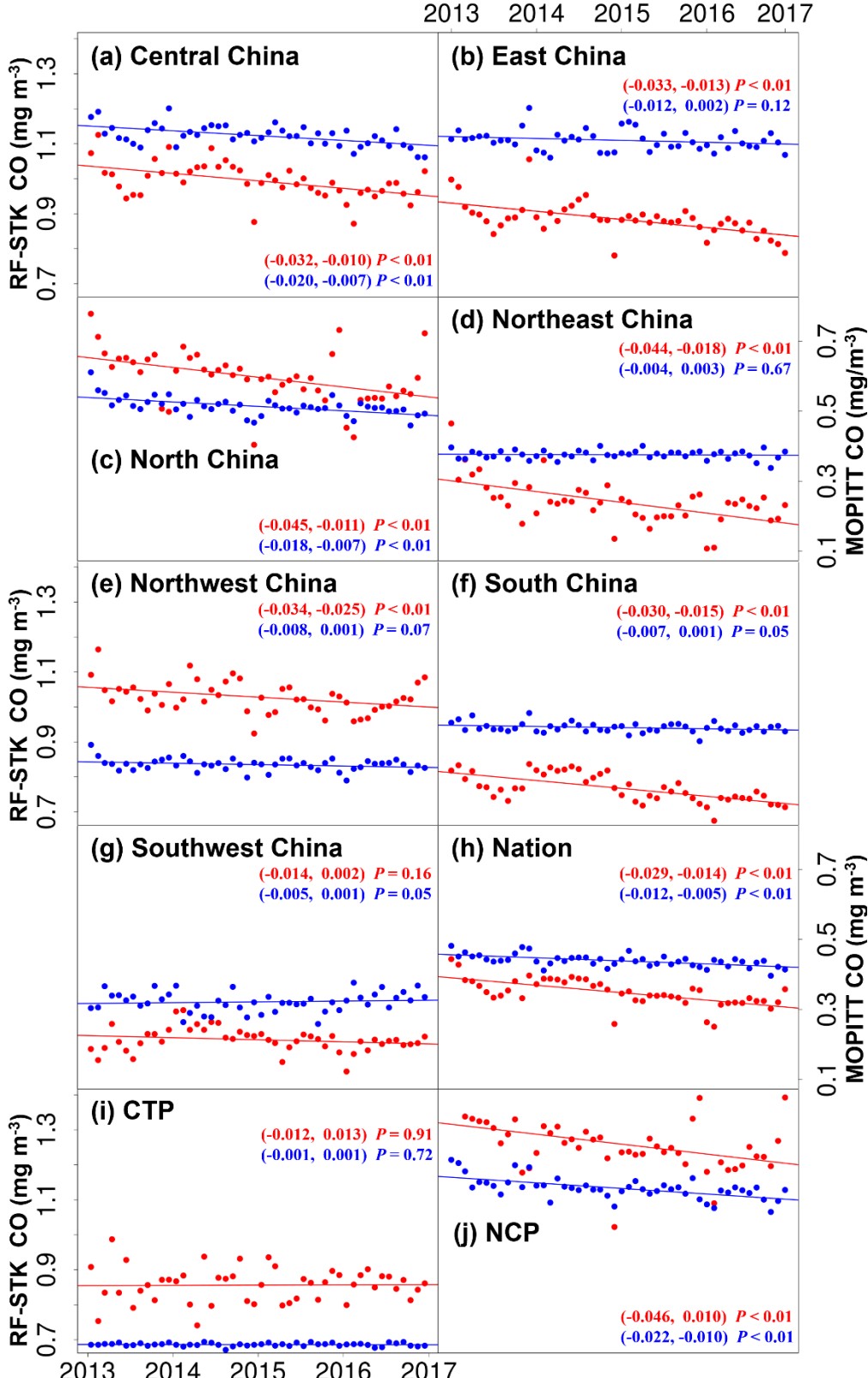

**Figure 9: Temporal trends of the population-weighted average ground-level CO concentrations (mg m⁻³) for (a) Central China, (b) East China, (c) North China, (d) Northeast China, (e) Northwest China, (f) South China, (g) Southwest China, (h) the whole nation, (i) the Central Tibetan Plateau (CTP), and (j) the North China Plain (NCP) during 2013-2016 based on the RF-STK predictions (red solid lines) and the MOPITT retrievals (blue solid lines). The points in different colors represent the deseasonalized monthly averages for deriving the corresponding trend lines. The 95% confidence intervals of the trends are in parentheses (mg m⁻³ per year) followed by the P values.**

