# Peer review of "Estimating ground-level CO concentrations across China based on national monitoring network and MOPITT: Potentially overlooked CO hotspots in the Tibetan Plateau"

_Atmospheric Chemistry and Physics, 2019_

## Referee Comment (RC1) · Anonymous Referee #1 · 14 May 2019

Reviewer's Comments on 'Estimating ground-level CO concentrations across China based on national monitoring network and MOPITT: Potentially overlooked CO hotspots in the Tibetan Plateau' by Liu et al.

General Comments

This manuscript reports an analysis of ground-level CO concentrations over China based on both measurements from a network of in-situ CO monitoring stations and the MOPITT satellite CO dataset. The writing is generally clear and the figures are of

good quality. However, while the subject matter is generally consistent with the focus of ACP, there are a number of important issues which are inadequately addressed in the manuscript. Revisions would be likely to involve considerable additional effort. Below are the major issues that I see, listed from the most general to most specific.

1. The overall goal of this research is not really clear. Shouldn't this paper attempt to show whether MOPITT data are or are not useful (combined with the surface in-situ network) for estimating surface-level CO concentrations over China? That question does not really seem answered by this study. Or, if the focus is on the RF-STK model and surface monitoring network, are the MOPITT data really even necessary for this research?

2. The sensitivity of MOPITT TIR-NIR surface-level CO retrievals is highly variable and should be analyzed more deeply. For example, averaging kernels should be presented for different regions (and perhaps different seasons) in order to anticipate situations where MOPITT surface-level retrievals should be useful versus situations where the retrievals will be strongly weighted by the a priori. For example, it would not be surprising if MOPITT averaging kernels over the Tibetan Plateau were generally weak (which is often true for mountainous regions), implying a strong dependence on the a priori.

3. A useful study would be to compare results for two experiments: one where the actual MOPITT retrieved surface-level CO is used in the analysis and a second experiment where the MOPITT a priori surface-level CO is used instead. Comparisons of the results for these two experiments with data from the monitoring network should reveal whether the MOPITT retrievals include additional useful information beyond the a priori.

4. As described in the MOPITT V7 validation paper, bias drift is a known issue for the MOPITT V7 TIR-NIR product. At the surface, bias drift appears to be approximately -0.7% per year, which is significant compared to actual CO trends. This artifact of the MOPITT data should be recognized and somehow represented in the data analysis.

5. Although MOPITT surface-level CO concentrations are reported in terms of volume mixing ratio (ppb), the manuscript consistently analyzes CO concentrations in terms of density (mg/m3). This issue is not discussed at all, although it is implied in several places that a single conversion factor is used to convert from ppb to mg/m3. Since CO density will decrease as the pressure decreases, and surface pressure varies considerably over China, a single conversion factor from ppb to mg/m3 is not appropriate.

6. The methods used for filtering and 'gap-filling' the MOPITT data (mentioned in Section 2.1) should be discussed in more detail.

7. For readers who are not familiar with machine learning methods (including myself), a more basic description of the RF-STK model in Section 2.3 would be helpful.

---

## Referee Comment (RC2) · Anonymous Referee #2 · 22 May 2019

It is important to understand the recent trends of ground-level CO concentrations over China for the accurate prediction of air quality. In this study, ground-based and satellite measurements of CO and a state-of-the-art modeling tool have been used to understand the recent trends of CO. I think this study has a high potential of becoming a reference for broad scientific community and for policy makers. However, I was somewhat overwhelmed by the amount of information presented throughout the manuscript with too little explanation. I think there is still room for improvement and below are the suggestions for the authors may want to take into consideration.

[Figure]

General Comments

- More background and historical context will be helpful in introduction. What is overall CO trend in China? How were the ground-level CO concentrations measured and used over China in the past? Is the data available in public? What are the short-comings of the in-situ measurements, satellite measurements and modeling studies? The goal of this study needs to be emphasized in introduction more clearly. Proper citations are needed for all the background information, specifically over China. Have there been similar studies like this?

- Why was the 2013-2016 period chosen? Is this period long enough to provide us reliable trend analysis? Wasn't the air pollution over China more severe before 2013?

- Since this study is focused on a smaller region, the importance of higher spatial and temporal resolution measurements and also higher resolution model should be mentioned as well. Considering MOPITT's large footprint (22x22 km2), is MOPITT the best fit for this type of study?

- I think there are many nice figures and tables included in the manuscript and supplement material. I would recommend reorganizing the figures with consistency. I find myself going back and forth the manuscript and supplement material trying to find the figures. I would also recommend to spend more time on describing figures and tables. Each figure contains more information than just being cited in the parenthesis.

- The results are presented here in the form of numbers and tables, which might give a quantitative information. However, it is somewhat challenging to see what the scientific messages are. I would recommend including tables only when is absolutely necessary. Figures are easier to understand otherwise.

- Section 3.3 and 3.4 contain mainly technical information and the figure numbers do not seem to have any particular order. Rewriting those sections with more explanation will help.

[Figure]

Specific Comments

P2, L12 – It is not clear the meaning of 'overlooked' here.

P2, L25 – A reference or more information needed.

P2, L27 – CTM -> CTMs

P2, L30-31 – Any references for this?

P3, L3-18 – More current references for all the satellite instruments are needed here.

P3, L10 – Deeter et al. (2014, 2017) should be included here.

P6, L18-19 – I wonder what is causing the sparse coverage over China. Also, how much coverage is considered to be enough or limiting here?

P6, L26-30 – For clarity, the authors need to include citations or data sources here. Also, how does the spatial coverage affect the bias and uncertainties?

P7, L4 – Why is the correlation coefficient higher in winter? What does the seasonal dependency in the correlation coefficients mean?

P7, L7-11 – For the MOPITT retrievals over the Tibetan Plateau, the authors might want to contact the MOPITT science team and seek for advice. Including the latest development in their retrieval methods will be useful here.

P7, L39 – Table S6 has so much information and it is not explained in the text at all.

P10, L1 – Explain what 'importance of coal consumption' specifically means and how is related to CO trend. Do people use less coal than before? Is combustion efficiency improving? Does this have a seasonal dependency?

P10, L6-7 – Why are the trends estimated by MOPITT lower?

P10, L10 – 'The refined RF-STK predictions that assimilates the MOPITT-CO with ground-level CO observations provide more solid information for decision making.' I

think this sentence is very important and should be in introduction.

P11, L29 – 'such as refining the prior status assigned to the overlooked hotspots in the Central Tibetan Plateau.'- I wonder how the results in this study can be utilized in improving MOPITT retrievals?

Figures and Tables

I am wondering how the figures in the manuscript and supplement material are divided. It seems like the figure descriptions in the text has no particular order.

Figure 1 – What is Heihe-Tengchong line? And what is the purpose of showing here? I do not see any relevance of inserting the South China Sea map as we are only considering the ground measurements stations here. I recommend removing the inserted map.

Table 1 – Higher correlation coefficients (> 0.9) can be marked as bold or shaded numbers for better visibility.

Table 2 can be replaced by bar-graphs, if it's possible. This applies to other tables included in supplements.

Figure S1 – How is the seasonal coverage calculated?

Figure S2 – Standard deviation (uncertainty) can be added here.

Figure S10 – What is partial dependence plot? Also, what are the x and y axes on this plot?

―――――――――――――――――

---

## Author Comment (AC1) · 31 Jul 2019

**Response to Anonymous Referee #1:**

This manuscript reports an analysis of ground-level CO concentrations over China based on both measurements from a network of in-situ CO monitoring stations and the MOPITT satellite CO dataset. The writing is generally clear and the figures are of good quality. However, while the subject matter is generally consistent with the focus of ACP, there are a number of important issues which are inadequately addressed in the manuscript. Revisions would be likely to involve considerable additional effort. Below are the major issues that I see, listed from the most general to most specific.

**Response:** We highly appreciate this referee's comments and have made a lot of effort to revise this manuscript. Please see the point-by-point responses below. Please note that the all numbers of pages and lines in the responses refer to the revised manuscript. The revised contents are also presented at the end of each response in quotes.

1. The overall goal of this research is not really clear. Shouldn't this paper attempt to show whether MOPITT data are or are not useful (combined with the surface in-situ network) for estimating surface-level CO concentrations over China? That question does not really seem answered by this study. Or, if the focus is on the RF-STK model and surface monitoring network, are the MOPITT data really even necessary for this research?

**Response:** The overall goal of this research is to estimate the spatiotemporal distribution of ground-level CO concentrations across China. To achieve this goal, we refined the hybrid random forest and spatiotemporal kriging (RF-STK) model, which assimilated the data from the national monitoring network and the MOPITT surface CO retrievals. In this manuscript, we compared the RF-STK predictions with the MOPITT retrievals, which might make the referee feel that the goal of this study is "to show whether MOPITT data are or are not useful (combined with the surface in-situ network) for estimating surface-level CO concentrations over China". Our purpose of comparing the RF-STK predictions with the MOPITT retrievals is to emphasize the improved CO estimations through the data assimilation. We have revised the manuscript thoroughly to clarify the overall goal of this research. Please see Page 3, Lines 32-38; Page 4, Lines 1-6.

The variable importance result for the RF-STK model suggests the high importance of the MOPITT data for estimating the spatiotemporal distributions of ground-level CO (Fig. 6). In the revision, we have conducted additional experiments to evaluate whether the MOPITT data are necessary for this research. Specifically, we compared the predictive performance of the RF-STK models with/without the MOPITT data by using two cross-validation strategies, including the site- and region-based cross-validation. The site-based cross-validation is a commonly used strategy but tends to overestimate the predictive performance given the fact of clustered distribution of monitoring sites. Therefore, we also employed the region-based cross-validation strategy by following the concept of cluster-based cross-validation, which was proposed to evaluate the importance of satellite retrievals for estimating surface $NO_2$ concentrations (Young et al., 2016).

The results of these new experiments show that the MOPITT data are necessary to achieve better predictive performance (Table 2). In the site-based cross-validation, removing MOPITT

data from the RF-STK model caused a slight decrease in the predictive performance ($R^2$ decreased from 0.51 to 0.49, and RMSE increased from 0.54 to 0.58 mg m$^{-3}$). In the region-based cross-validation, the predictive performance decreased considerably after excluding the MOPITT data ($R^2$ decreased from 0.45 to 0.32, and RMSE increased from 0.61 to 0.69 mg m$^{-3}$). Therefore, the MOPITT data are crucial to improving the predictive performance of the RF-STK model especially for the areas without monitoring sites nearby. In the revision, we have added the results of these new experiments and emphasized the importance of the MOPITT data for this research. Please see Page 6, Lines 15-26; Page 9, Lines 11-12; Page 9, Lines 37-38; Page 10, Lines 1-6.

[revised manuscript text omitted]

2. The sensitivity of MOPITT TIR-NIR surface-level CO retrievals is highly variable and should be analyzed more deeply. For example, averaging kernels should be presented for different regions (and perhaps different seasons) in order to anticipate situations where MOPITT surface-level retrievals should be useful versus situations where the retrievals will be strongly weighted by the a priori. For example, it would not be surprising if MOPITT averaging kernels over the Tibetan Plateau were generally weak (which is often true for mountainous regions), implying a strong dependence on the a priori.

**Response:** Suggestion is taken. The averaging kernels for the MOPITT TIR-NIR CO retrievals have been analyzed to show the spatiotemporal variations in the dependence of the MOPITT surface CO retrievals on the a priori information. For each averaging kernel (a matrix), the sum of the elements in the row associated with the surface layer of the CO profile (hereafter referred to as the row-sum value) is an integrative measure of the dependence (Deeter, 2017). A small row-sum value indicates strong dependence of the MOPITT retrieval on the a priori. Among the seven geographic regions of China, the average row-sum values during 2013-2016 were the highest in East China and the lowest in Northeast China. Seasonally, the national average row-sum values were the highest in fall and the lowest in summer/winter. It is noteworthy that the row-sum values were apparently lower in the Central Tibetan Plateau (CTP) than the North China Plain (NCP), suggesting stronger dependence of the MOPITT retrievals on the a priori information, i.e., lower sensitivity, for CTP. In the revision, we have presented the spatiotemporal distributions of the averaging kernels related to the ground-level CO retrievals. The description of the averaging kernel has been added to the supplementary data. Please see Page 5, Lines 3-9; Page 8, Lines 10-17; Page 11, Lines 37-38; Page 12, Lines 1-3; Figure S4; and section S.2 in the Supplementary Data.

Page 5, Lines 3-9.
"In order to evaluate the dependence of the MOPITT surface retrievals on the a priori information, we also extracted the averaging kernels and the a priori information from the MOPITT product. For each averaging kernel (a matrix), the sum of the elements in the row associated with the surface layer of the CO profile (hereafter referred to as the row-sum value) measures the overall dependence of the MOPITT surface CO retrievals on the a priori information (Deeter, 2017). A small row-sum value indicates strong dependence of the MOPITT retrieval on the a priori information, i.e., low sensitivity of the actual MOPITT retrieval. Please refer to Section S.2 in the Supplementary Data for the explanation of the averaging kernels."

Page 8, Lines 10-17.

"The spatiotemporal distributions of the row-sum values of the averaging kernels demonstrate that the dependence of the MOPITT retrievals on the a priori varied widely (Fig. S4). Among the seven geographic regions of China, the average row-sum values during 2013-2016 were the highest in East China and the lowest in Northeast China. Seasonally, the national average row-sum values were the highest in fall and the lowest in summer/winter. The row-sum values were lower in CTP than NCP, suggesting a stronger dependence of the MOPITT retrievals on the a priori in CTP than NCP. The variations in the sensitivity of the MOPITT retrievals could result from various sources, such as the CO amounts and the diurnal temperature differences (Deeter, 2003; Deeter, 2007; Worden et al., 2013b)."

Page 11, Lines 37-38; Page 12, Lines 1-3.
"The trend underestimation by MOPITT-CO might be largely due to the setting that the a priori information was the same across the years (Dekker et al., 2017). We found that the trend underestimation tended to be more severe for the regions with weaker averaging kernels (Figs. 9 and S5), which was analogous to the phenomenon that the predictions made by the RF-STK model with the a priori information exhibited a slower decreasing rate (-2.06% per year) than the model with MOPITT-CO (Fig. S15)."

Page 2 in the Supplementary Data
"**S.2 Averaging kernel**
The averaging kernel (matrix $A$) adjusts the weights of the "true" state (vector $x$) and the a priori (vector $x_a$) in deriving the MOPITT CO retrievals (vector $\hat{x}$) (Deeter, 2003; Rodgers, 2000).
$$\hat{x} \approx Ax + (I - A)x_a \tag{5}$$
where $I$ is the identity matrix. Each row of $A$ corresponds to a vertical layer of the CO profile, and the sum of a row shows the overall dependence of the MOPITT CO retrieval at that layer on the a priori information. A small row-sum value indicates strong dependence on the a priori information."

[Figure]

**Figure S4:** Seasonal means of the averaging-kernel row-sum values associated with the MOPITT retrieved surface CO for (a) spring, (b) summer, (c) fall, and (d) winter during 2013-2016 across China. Small row-sum values indicate strong dependence of the MOPITT retrievals on the a priori information. Please refer to "S.2 Averaging kernel" for more explanation.

References for this response

Deeter, M. N., Emmons, L. K., Francis, G. L., Edwards, D. P., Gille, J. C., Warner, J. X., Khattatov, B., Ziskin, D., Lamarque, J. F., Ho, S. P., Yudin, V., Attié, J. L., Packman, D., Chen, J., Mao, D., and Drummond, J. R.: Operational carbon monoxide retrieval algorithm and selected results for the MOPITT instrument, J. Geophys. Res-Atmos., 108, 10.1029/2002jd003186, 2003.

Deeter, M. N.: A new satellite retrieval method for precipitable water vapor over land and ocean, Geophys. Res. Lett., 34, 10.1029/2006gl028019, 2007.

Deeter, M. N.: Measurements of Pollution in the Troposphere (MOPITT) Version 7 Product User's Guide, varilable at: https://www2.acom.ucar.edu/sites/default/files/mopitt/v7_users_guide_201707.pdf, 2017.

Worden, H. M., Deeter, M. N., Frankenberg, C., George, M., Nichitiu, F., Worden, J., Aben, I., Bowman, K. W., Clerbaux, C., Coheur, P. F., de Laat, A. T. J., Detweiler, R., Drummond, J. R., Edwards, D. P., Gille, J. C., Hurtmans, D., Luo, M., Martínez-Alonso, S., Massie, S., Pfister, G., and Warner, J. X.: Decadal record of satellite carbon monoxide observations, Atmos. Chem. Phys., 13, 837-850, 10.5194/acp-13-837-2013, 2013b.

Rodgers, C. D.: Inverse Methods for Atmospheric Sounding, Theory and Practice, World Scientific, 2000.

3.  A useful study would be to compare results for two experiments: one where the actual MOPITT retrieved surface-level CO is used in the analysis and a second experiment where the MOPITT a priori surface-level CO is used instead. Comparisons of the results for these two experiments with data from the monitoring network should reveal whether the MOPITT retrievals include additional useful information beyond the a priori.

**Response:** The suggestion is taken. A new experiment has been conducted by using the MOPITT a priori surface-level CO in the analysis. The results of this new experiment and the existing one that uses the actual MOPITT retrieved surface-level CO are compared with the ground-level CO observations from the monitoring network. The purpose of performing this new experiment is to evaluate whether the actual MOPITT retrievals include additional useful information beyond the a priori. The main results are summarized below.

- While the actual MOPITT retrievals correlate well with the a priori ($r = 0.89$), the ground-level CO observations from the monitoring network exhibit stronger correlation with the actual retrievals ($r = 0.37$) than the a priori ($r = 0.30$; Table 1).
- On the basis of the site-based cross-validation results, the RF-STK model with the actual MOPITT retrievals shows better predictive performance than the model with the a priori, with $R^2 = 0.51$ vs. 0.49 and slope = 0.64 vs. 0.60 (Table S4).
- As the MOPITT a priori remains the same for each year (Dekker et al., 2017), the interannual variation in the ground-level CO concentrations tends to be underestimated by the RF-STK model using the a priori (Figure S15). The observations from the monitoring network show that the national annual average ground-level CO concentrations decreased by 6.4% from 2013 to 2016. The predictions made by the RF-STK model using the actual MOPITT retrievals show a decreasing rate of 6.9% from 2013 to 2016, compared to 5.2% estimated by the RF-STK model using the a priori.

In summary, the actual MOPITT retrievals indeed include additional useful information beyond the a priori. These above results have been added to the revised manuscript. Please see Page 8, Lines 6-10; Page 9, Lines 11-12; Page 9, Lines 37-38; Page 10, Line 1; Page 11, Lines 37-38; Page 12, Lines 1-3; Figure S15, Table 1.

Page 8, Lines 6-10.
"The spatiotemporal pattern of the MOPITT-CO was generally consistent with that of the ground-level CO observations in China, with $r = 0.43$ for the multiyear averages and $r = 0.37$ for the daily values during 2013-2016 (Table 1). The correlation between the a priori and the ground-level observations was weaker, with $r = 0.34$ for the multiyear averages and $r = 0.30$ for the daily values, suggesting that the MOPITT retrievals provided more information on the ground-level CO distributions than the a priori."

Page 9, Lines 11-12; Page 9, Lines 37-38; Page 10, Line 1.

"On the basis of the site-based cross-validation results, the RF-STK model showed reasonable performance in predicting the daily ground-level CO concentrations, with $R^2$=0.51, RMSE=0.54 mg m$^{-3}$, and slope=0.64 (Fig. 5).

It is noteworthy that the RF-STK model with MOPITT-CO was superior to the model without MOPITT-CO ($R^2$=0.49, RMSE=0.58 mg m$^{-3}$, and slope=0.60) and the model with the a priori information ($R^2$=0.49, RMSE=0.57 mg m$^{-3}$, and slope=0.60) based on the site-based cross-validation results (Tables 2 and S4)."

Page 11, Lines 37-38; Page 12, Lines 1-3.
"The trend underestimation by MOPITT-CO might be largely due to the setting that the a priori information was the same across the years (Dekker et al., 2017). We found that the trend underestimation tended to be more severe for the regions with weaker averaging kernels (Figs. 9 and S5), which was analogous to the phenomenon that the predictions made by the RF-STK model with the a priori information exhibited a slower decreasing rate (-2.06% per year) than the model with MOPITT-CO (Fig. S15)."

**Table 1.** Correlations among the ground observations, MOPITT-CO, and the RF-STK predictions (Pearson correlation coefficients).

| Region/Dataset | Pair[a] | Daily | Monthly | Seasonal | Annual | Spatial[b] |
|---|---|---|---|---|---|---|
| Nation | O-M | 0.37 | 0.40 | 0.45 | 0.44 | 0.43 |
| | O-P | **0.95** | **0.97** | **0.97** | **0.97** | **0.98** |
| | P-M | 0.09 | 0.1 | 0.1 | 0.09 | 0.13 |
| Central Tibetan | O-M | -0.03 | -0.04 | 0.11 | -0.12 | -0.12 |
| Plateau (CTP)[c] | O-P | **0.91** | **0.92** | **0.93** | **0.96** | **1** |
| | P-M | -0.04 | -0.04 | -0.06 | -0.09 | -0.12 |
| North China | O-M | 0.35 | 0.36 | 0.40 | 0.30 | 0.20 |
| Plain (NCP)[c] | O-P | **0.95** | **0.97** | **0.98** | **0.97** | **0.98** |
| | P-M | 0.35 | 0.40 | 0.47 | 0.52 | 0.58 |
| X1_PRI[d] | O-M | 0.30 | 0.32 | 0.38 | 0.34 | 0.34 |
| X1_TS[d] | O-M | 0.39 | 0.47 | 0.49 | 0.44 | 0.42 |
| X2[d] | O-M | 0.37 | 0.39 | 0.45 | 0.42 | 0.40 |

[a] O: ground-level CO observations; M: MOPITT-CO; P: predictions made by the RF-STK model; the correlation coefficients higher than 0.90 are in bold.
[b] Multiyear averages during 2013-2016.
[c] Please refer to Fig. 1 for the locations of CTP and NCP.
[d] X1_PRI: nationwide a priori for MOPITT-CO; X1_TS: nationwide MOPITT-CO processed with the temporal and spatial convolution; X2: nationwide nighttime MOPITT-CO, and all the other MOPITT-CO data refer to daytime retrievals.

**Table S4.** Comparisons of the RF and RF-STK models in predicting daily ground-level CO concentrations across China during 2013-2016 based on the 10-fold cross-validation.

| Metric[a] | RF$_r$[b] | RF[b] | RF$_{rw}$[b] | RF$_w$[b] | RF$_w$-STK[b] | RF$_{rw}$-STK[bc] | RF$_{rw}$-STK[b] |
|---|---|---|---|---|---|---|---|

| | | | | | | | |
|---|---|---|---|---|---|---|---|
| $R^2$ | 0.56 | 0.53 | 0.54 | 0.53 | 0.49 | 0.49 | 0.51 |
| Slope | 0.60 | 0.55 | 0.57 | 0.55 | 0.63 | 0.60 | 0.64 |
| RMSE | 0.50 | 0.52 | 0.51 | 0.52 | 0.55 | 0.57 | 0.54 |
| RPE | 46.1% | 48.0% | 47.1% | 48.1% | 51.0% | 53.3% | 50.4% |
| MFB | 0.083 | -0.013 | -0.008 | -0.013 | -0.030 | 0.064 | -0.022 |
| MFE | 0.31 | 0.31 | 0.31 | 0.31 | 0.36 | 0.37 | 0.35 |
| MNB | 0.90 | 0.64 | 0.66 | 0.64 | 0.68 | 0.74 | 0.70 |
| MNE | 1.09 | 0.90 | 0.91 | 0.90 | 0.97 | 1.0 | 0.98 |

[a] $R^2$: coefficient of determination; RMSE: root mean square error (mg m$^{-3}$); RPE: relative prediction error; MFB: mean fractional bias; MFE: mean fractional error; MNB: mean normalized bias; MNE: mean normalized error.

[b] RF: random forest; STK: spatiotemporal kriging. Subscript r indicates a reduced model through variable selection, and subscript w means that the training samples were inversely weighted by the associated population densities. The CO concentrations were log-transformed to train all the models except for RF$_r$ which was trained with the CO concentrations at native scale.

[c] This RF-STK model was developed with the a priori information rather than the MOPITT retrievals.

[Figure]

**Figure S15:** Temporal trends of the population-weighted average ground-level CO concentrations (mg m$^{-3}$) for China during 2013-2016 based on the actual MOPITT retrieved surface CO (blue solid line), the MOPITT a priori surface CO (purple solid line), the predictions made by the RF-STK model using the actual MOPITT retrieved surface CO (red solid line), and the predictions made by the RF-STK model using the MOPITT a priori surface CO (black solid line). The points in different colors represent the deseasonalized monthly averages for deriving the corresponding trend lines. The 95% confidence intervals of the trends are in parentheses followed by the $P$ values.

4. As described in the MOPITT V7 validation paper, bias drift is a known issue for the MOPITT V7 TIR-NIR product. At the surface, bias drift appears to be approximately -0.7% per year, which is significant compared to actual CO trends. This artifact of the MOPITT data should be recognized and somehow represented in the data analysis.

**Response:** Suggestion is taken. The retrieval bias drift at the surface for the MOPITT V7 TIR/NIR product was found to be approximately -0.69% per year based on the flask measurements performed by the National Oceanic and Atmospheric Administration (Deeter et al., 2017). We assumed that the bias values were also applicable to China and consequently adjusted the MOPITT retrieved surface CO values. We then compared the temporal trends of the original MOPITT retrievals, the adjusted MOPITT retrievals, and the observations from the monitoring network. These three datasets were merged into a complete dataset through spatiotemporal matching, and the rows with missing values were excluded to make the trends derived from these three datasets strictly comparable.

In order to be consistent with the form of bias drift (% per year), the relative temporal trends (% per year) were estimated by fitting exponential equations to the deseasonalized monthly averages. The temporal trend of the observation data from the monitoring network was -5.53% per year. The temporal trend of the adjusted MOPITT retrievals was -3.51% per year, compared to -4.17% per year for the original retrievals. As we can see, the trend underestimation by the MOPITT retrieval was more severe after the "bias correction". Therefore, we suspected that the bias values found in that validation paper (Deeter et al., 2017) might not be applicable to China, since the bias drift values showed considerable spatial variation (Buchholz et al., 2017). More efforts are therefore needed to evaluate the bias of the MOPITT retrievals for China. In the revision, we have discussed the issue of bias drift for the MOPITT V7 TIR/NIR product. Please see Page 12, Lines 4-12.

Page 12, Lines 4-12.
"The issue of bias drift for the MOPITT retrievals, which could result from long-term instrumental degradation (Deeter et al., 2017), should also be considered in the trend analyses. The bias drift for MOPITT-CO was found to be approximately -0.69% per year based on the flask measurements performed by the National Oceanic and Atmospheric Administration (Deeter et al., 2017). It is noteworthy that the extents of bias drift were of considerable spatial variation (Buchholz et al., 2017). For the present study, if the MOPITT-CO data were "corrected" by the bias drift of -0.69%, the relative change rate of MPW would become lower (-1.31% per year), and the trend underestimation by the MOPITT would be more severe (Fig. S16). Accurate information on the temporal trends of CO is essential for air quality management, and more efforts are thus required to improve the data quality of CO measurements."

[Figure]

**Figure S16:** Temporal trends of the population-weighted average ground-level CO concentrations (mg m$^{-3}$) for China during 2013-2016 based on the actual (blue solid line) and the bias-adjusted (green solid line) MOPITT retrieved surface CO, as well as the predictions made by the RF-STK model using the actual MOPITT retrieved surface CO (red solid line). The bias correction was carried out according to the mean bias drift of -0.69% per year reported in the previous study (Deeter et al., 2017). The points in different colors represent the deseasonalized monthly averages for deriving the corresponding trend lines. The 95% confidence intervals of the trends (mg m$^{-3}$ per year) are in parentheses followed by the *P* values.

References for this response
Buchholz, R. R., Deeter, M. N., Worden, H. M., Gille, J., Edwards, D. P., Hannigan, J. W., Jones, N. B., Paton-Walsh, C., Griffith, D. W. T., Smale, D., Robinson, J., Strong, K., Conway, S., Sussmann, R., Hase, F., Blumenstock, T., Mahieu, E., and Langerock, B.: Validation of MOPITT carbon monoxide using ground-based Fourier transform infrared spectrometer data from NDACC, Atmos. Meas. Tech., 10, 1927-1956, 10.5194/amt-10-1927-2017, 2017.
Deeter, M. N., Edwards, D. P., Francis, G. L., Gille, J. C., Martínez-Alonso, S., Worden, H. M., and Sweeney, C.: A climate-scale satellite record for carbon monoxide: the MOPITT Version 7 product, Atmos. Meas. Tech., 10, 2533-2555, 10.5194/amt-10-2533-2017, 2017.

5. Although MOPITT surface-level CO concentrations are reported in terms of volume mixing ratio (ppb), the manuscript consistently analyzes CO concentrations in terms of density (mg/m$^3$). This issue is not discussed at all, although it is implied in several places that a single conversion factor is used to convert from ppb to mg/m$^3$. Since CO density will decrease as the pressure decreases, and surface pressure varies considerably over China, a single conversion factor from ppb to mg/m$^3$ is not appropriate.

**Response:** Suggestion is taken. In the revision, we have employed a commonly used equation to convert the unit of the MOPITT surface-level CO concentrations from ppb to mg m$^{-3}$, which takes into account the atmospheric pressure and temperature. This equation based on the ideal gas law shows that either higher pressure or temperature leads to a larger conversion factor from ppb to mg m$^{-3}$. Compared with the previous results based on a single conversion factor,

the new MOPITT-CO density (mg m$^{-3}$) became lower in low-pressure regions such as Tibetan Plateau and higher in high-pressure regions such as the North China Plain. The results of MOPITT surface-level CO have been updated throughout the revised manuscript. Please see Figures 3 and S5.

Page 4, Lines 31-35; Page 5, Lines 1-2.
"According to the ideal gas law, we converted the unit of MOPITT-CO data from ppb (the unit presented in the MOPITT product) to mg m$^{-3}$ in order to be comparable with the CO observations from the monitoring network:

$$C = B \cdot P \cdot M / (R \cdot T) \tag{1}$$

where $C$ is the CO concentration in the unit of mg m$^{-3}$, $B$ is the CO concentration in the unit of ppb, $P$ is the atmospheric pressure (atm), $M$ is the molecular weight of CO (mg mol$^{-1}$), $R$ is the gas constant (0.082 L atm mol$^{-1}$ K$^{-1}$), and $T$ is the atmospheric temperature (K). Note that the data of atmospheric pressure and temperature for the unit conversion are available in the MOPITT product."

[Figure]

**Figure 3:** Seasonal averages of the MOPITT retrieved surface CO concentrations (mg m$^{-3}$) in (a) spring, (b) summer, (c) fall, and (d) winter during 2013-2016 across China.

[Figure]

**Figure S5:** Annual averages of ground-level CO concentrations retrieved by the MOPITT for (a) 2013, (b) 2014, (c) 2015, and (d) 2016.

6. The methods used for filtering and 'gap-filling' the MOPITT data (mentioned in Section 2.1) should be discussed in more detail.

**Response:** Suggestion is taken. The temporal and spatial convolution was used for filtering and "gap-filling" the MOPITT data. In the first step, the temporal convolution with a 1-dimensional Gaussian kernel was used to process the MOPITT retrievals for each grid cell. In the second step, the spatial convolution with a 2-dimensional Gaussian kernel was employed to processed the output from the first step day by day. After the convolution processing, the coverage rate of MOPITT data elevated from 3.5% to 100%, while the correlation between the daily MOPITT data and the daily observations from the monitoring network remained stable ($r$ = 0.37 vs. 0.39; Table 1). The processed full-coverage MOPITT data were essential for estimating the daily CO concentrations across China. In the revision, we have added more descriptions of the temporal and spatial convolution for processing the MOPITT data. Please see Page 4, Lines 26-30; Section S.1 in the supplementary Data.

Page 4, Lines 26-30.
"Through the temporal and spatial convolution with Gaussian kernels (Goodfellow et al., 2016), the MOPITT noise was filtered and the data gaps were filled, which were then resampled to the 0.1° grid. Briefly, the MOPITT-CO data for each grid cell were first processed with the temporal convolution, which were then processed with the spatial convolution day by day. Please refer to Section S.1 in the Supplementary Data for the mathematical equations."

Page 2 in the Supplementary Data

**"S.1 Temporal and spatial convolution**

The data of MOPITT retrieved surface CO (MOPITT-CO) are processed with the temporal and spatial convolution to filter noises and fill data gaps. In the first step, the temporal convolution with a 1-dimensioanl Gaussian kernel is employed to process the MOPITT-CO data for each grid cell:

$$M_T(t_0) = \sum_t [M(t) \cdot W_T(t_0 - t)] / \sum_t W_T(t_0 - t) \tag{1}$$

where $M_T(t_0)$ is the output value on day $t_0$ processed by the temporal convolution, $M(t)$ is the original MOPITT-CO value on day $t$, and $W_T(t_0 - t)$ is the weighting factor determined by the 1-dimensional Gaussian function:

$$W_T(t_0 - t) = exp[-(t_0 - t)^2/(2\sigma_T^2)] \tag{2}$$

where the standard deviation ($\sigma_T$) is set to 60 according to the sensitivity analysis on the completeness and smoothness of the processed data.

In the second step, the spatial convolution with a 2-dimensional Gaussian kernel is employed to process the output from the previous step day by day:

$$M_{TS}(x_0, y_0) = \sum_{x,y} [M_T(x, y) \cdot W_S(x_0 - x, y_0 - y)] / \sum_{x,y} W_S(x_0 - x, y_0 - y) \tag{3}$$

where $M_{TS}(x_0, y_0)$ is the output value for cell $(x_0, y_0)$ processed by the spatial convolution, $M_T(x, y)$ is the processed MOPITT-CO value from the first step for cell $(x, y)$, and $W_S(x_0 - x, y_0 - y)$ is the weighting factor determined by the 2-dimensional Gaussian function:

$$W_S(x_0 - x, y_0 - y) = exp\{-[(x_0 - x)^2 + (y_0 - y)^2)]/2\sigma_S^2\} \tag{4}$$

where the standard deviation ($\sigma_S$) is set to 0.1 according to the sensitivity analysis on the completeness and smoothness of the processed data."

7. For readers who are not familiar with machine learning methods (including myself), a more basic description of the RF-STK model in Section 2.3 would be helpful.

**Response:** Suggestion is taken. The RF-STK model is a hybrid model of random forest and spatiotemporal kriging, which was proposed to estimate the spatiotemporal distribution of daily ambient $NO_2$ concentrations across China (Zhan, et al., 2018). In the present study, we refined the original RF-STK model by inversely weighting the training samples with their surrounding population density to alleviate the effects of sampling bias towards populous areas. In the revision, we have added a more basic description of the refined RF-STK model to Section 2.3. Please see Page 5, Lines 11-24.

Page 5, Lines 11-24.

[revised manuscript text omitted]

The data of MOPITT retrieved surface CO (MOPITT-CO) are processed with the temporal and spatial convolution to filter noises and fill data gaps. In the first step, the temporal convolution with a 1-dimensioanl Gaussian kernel is employed to process the MOPITT-CO data for each grid cell:

$$M_T(t_0) = \sum_t [M(t) \cdot W_T(t_0 - t)] / \sum_t W_T(t_0 - t) \tag{1}$$

where $M_T(t_0)$ is the output value on day $t_0$ processed by the temporal convolution, $M(t)$ is the original MOPITT-CO value on day $t$, and $W_T(t_0 - t)$ is the weighting factor determined by the 1-dimensional Gaussian function:

$$W_T(t_0 - t) = exp[-(t_0 - t)^2/(2\sigma_T^2)] \tag{2}$$

where the standard deviation ($\sigma_T$) is set to 60 according to the sensitivity analysis on the completeness and smoothness of the processed data.

In the second step, the spatial convolution with a 2-dimensional Gaussian kernel is employed to process the output from the previous step day by day:

$$M_{TS}(x_0, y_0) = \sum_{x,y} [M_T(x, y) \cdot W_S(x_0 - x, y_0 - y)] / \sum_{x,y} W_S(x_0 - x, y_0 - y) \tag{3}$$

where $M_{TS}(x_0, y_0)$ is the output value for cell $(x_0, y_0)$ processed by the spatial convolution, $M_T(x, y)$ is the processed MOPITT-CO value from the first step for cell $(x, y)$, and $W_S(x_0 - x, y_0 - y)$ is the weighting factor determined by the 2-dimensional Gaussian function:

$$W_S(x_0 - x, y_0 - y) = exp\{-[(x_0 - x)^2 + (y_0 - y)^2)]/2\sigma_S^2\} \tag{4}$$

where the standard deviation ($\sigma_S$) is set to 0.1 according to the sensitivity analysis on the completeness and smoothness of the processed data.

**S.2 Averaging kernel**

The averaging kernel (matrix $A$) adjusts the weights of the "true" state (vector $x$) and the a priori (vector $x_a$) in deriving the MOPITT CO retrievals (vector $\hat{x}$) (Deeter et al., 2003; Rodgers, 2000).

$$\hat{x} \approx Ax + (I - A)x_a \tag{5}$$

where $I$ is the identity matrix. Each row of $A$ corresponds to a vertical layer of the CO profile, and the sum of a row shows the overall dependence of the MOPITT CO retrieval at that layer on the a priori information. A small row-sum value indicates strong dependence on the a priori information.

**S.3 Algorithm of random forests (Breiman, 2001)**

For *tree* = 1 to *N* (e.g., 500 trees in this study):

◆ Randomly draw a sample from the training data with replacement through bootstrapping;

◆ A tree is grown from a single node, and the following steps are repeated until the minimum number of observations is present at each terminal node:

   ✧ Randomly select a subset of predictors to be considered at each split;

   ✧ Find the split that reduces the squared error the most;

Average the predictions made by all the decision trees as the output of the random forest.

**S.4 Environmental condition data**

◆ The daily weather conditions, including atmospheric pressure, air temperature, precipitation, evaporation, insolation duration, and wind speed, were obtained from 839 meteorological stations (CMA, 2017).

◆ The elevation data were retrieved from the Shuttle Radar Topography Mission (SRTM) database (Jarvis et al., 2016).

◆ The data of population density, road density, and land use were extracted from the Gridded Population of the World, the OpenStreetMap, and the GlobeLand30 databases, respectively (CIESIN, 2016; OSP, 2016; Jun et al., 2014).

◆ The daily planetary boundary height (PBLH) data were obtained from the Modern-Era Retrospective Analysis for Research and Application (GMAO, 2015).

◆ The Normalized Difference Vegetation Index (NDVI) data were retrieved from the Moderate Resolution Imaging Spectroradiometer (MODIS) satellite retrievals (Didan et al., 2015).

◆ The anthropogenic emission inventories were obtained from the Multi-resolution Emission Inventory for China (MEIC) database (Li et al., 2017). Due to the data availability, the emissions for 2013 and 2015 were linearly interpolated from the available emission data for 2012, 2014, and 2016.

**Table S1.** List of variable symbols and definitions.

| Symbol | Unit | Variable definition | Spatial[a] | Temporal[a] | Convolution[b] |
|---|---|---|---|---|---|
| MOPITT | molecule cm$^{-2}$ | MOPITT-retrieved CO surface mixing ratio | 0.25° | Day | Temporal and Spatial |
| DOY | - | Day of year | - | - | - |
| YEAR | - | Year | - | - | - |
| EVP | mm | Evaporation | Point | Day | - |
| PRE | mm | Precipitation | Point | Day | - |
| PRS | hPa | Atmospheric pressure | Point | Day | - |
| RHU | % | Relative humidity | Point | Day | - |
| SSD | hour | Sunshine duration | Point | Day | - |
| TEM | °C | Temperature | Point | Day | - |
| WIN | m s$^{-1}$ | Wind speed | Point | Day | - |
| PBLH | Km | Planetary boundary layer height | 0.625°×0.5° | Day | - |
| ELV | M | Elevation | 90 m | - | Spatial |
| NDVI | - | Normalized Difference Vegetation Index | 250 m | 8 Days | Spatial |
| POP | people km$^{-2}$ | Population density | 30" | - | Spatial |
| LU10 | % | Cultivated land area | 30 m | - | Spatial |
| LU20 | % | Forest area | 30 m | - | Spatial |
| LU30 | % | Grassland area | 30 m | - | Spatial |
| LU40 | % | Shrubland area | 30 m | - | Spatial |
| LU50 | % | Wetland area | 30 m | - | Spatial |
| LU60 | % | Waterbody area | 30 m | - | Spatial |
| LU80 | % | Artificial surface area | 30 m | - | Spatial |
| LU90 | % | Bareland area | 30 m | - | Spatial |
| LU100 | % | Permanent frozen land area | 30 m | - | Spatial |
| LU255 | % | Sea area | 30 m | - | Spatial |
| ROAD | Km grid$^{-1}$ | Road density | Polyline | - | Spatial |
| eBC | Mg grid$^{-1}$ | Emission of black carbon | 0.25° | Month | Spatial |
| eCO | Mg grid$^{-1}$ | Emission of CO | 0.25° | Month | Spatial |
| eCO2 | Mg grid$^{-1}$ | Emission of $CO_2$ | 0.25° | Month | Spatial |
| eNH3 | Mg grid$^{-1}$ | Emission of $NH_3$ | 0.25° | Month | Spatial |
| eNOx | Mg grid$^{-1}$ | Emission of $NO_2$ and NO | 0.25° | Month | Spatial |
| eOC | Mg grid$^{-1}$ | Emission of organic carbon | 0.25° | Month | Spatial |
| ePM25 | Mg grid$^{-1}$ | Emission of $PM_{2.5}$ | 0.25° | Month | Spatial |
| ePMcoar | Mg grid$^{-1}$ | Emission of PM-coarse | 0.25° | Month | Spatial |
| eSO2 | Mg grid$^{-1}$ | Emission of $SO_2$ | 0.25° | Month | Spatial |
| eVOC | Mg grid$^{-1}$ | Emission of VOC | 0.25° | Month | Spatial |

[a] Spatial or temporal resolution of raw data.

[b] Temporal: MOPITT is processed with the temporal and spatial convolution. Spatial: These variables have accompanying variables processed with the spatial convolution.

**Table S2.** Coverage rates of MOPITT-CO retrievals across China ($\mu \pm \sigma$; %)[a].

| Year(s) | Spring | Summer | Fall | Winter | Annual |
|---|---|---|---|---|---|
| 2013 | $3.2 \pm 2.1$ | $3.0 \pm 2.0$ | $4.5 \pm 2.6$ | $4.1 \pm 2.7$ | $3.7 \pm 0.6$ |
| 2014 | $3.1 \pm 2.1$ | $2.7 \pm 2.0$ | $4.3 \pm 2.5$ | $4.1 \pm 2.4$ | $3.6 \pm 0.6$ |
| 2015 | $3.0 \pm 2.0$ | $3.0 \pm 2.1$ | $4.0 \pm 2.4$ | $3.8 \pm 2.4$ | $3.4 \pm 0.6$ |
| 2016 | $3.0 \pm 2.1$ | $3.0 \pm 2.0$ | $4.0 \pm 2.7$ | $3.6 \pm 2.3$ | $3.4 \pm 0.6$ |
| 2013-2016 | $3.1 \pm 1.6$ | $2.9 \pm 1.5$ | $4.2 \pm 1.9$ | $3.9 \pm 2.0$ | $3.5 \pm 0.5$ |

[a] $\sigma$ stands for the spatial variation. Please refer to Fig. S2 for the coverage maps.

**Table S3.** Correlations between the daily CO observations from the monitoring network and the MOPITT surface retrievals for China during 2013-2016.

| Region[a] | Spring | Summer | Fall | Winter | 2013 | 2014 | 2015 | 2016 | 2013-2016 |
|---|---|---|---|---|---|---|---|---|---|
| Central China | 0.08 | 0.18 | 0.21 | 0.27 | 0.59 | 0.22 | 0.32 | 0.24 | 0.30 |
| East China | 0.12 | 0.24 | 0.36 | 0.37 | 0.46 | 0.34 | 0.48 | 0.40 | 0.43 |
| North China | 0.30 | 0.23 | 0.31 | 0.42 | 0.36 | 0.31 | 0.39 | 0.38 | 0.37 |
| Northeast China | 0.28 | 0.34 | 0.38 | 0.30 | 0.28 | 0.41 | 0.39 | 0.38 | 0.39 |
| Northwest China | 0.31 | 0.12 | 0.27 | 0.31 | 0.52 | 0.39 | 0.35 | 0.31 | 0.38 |
| South China | 0.49 | 0.53 | 0.59 | 0.52 | 0.65 | 0.56 | 0.55 | 0.57 | 0.58 |
| Southwest China | 0.10 | 0.12 | 0.25 | 0.20 | 0.14 | -0.02 | 0.17 | 0.22 | 0.17 |
| Nation | 0.22 | 0.31 | 0.32 | 0.34 | 0.42 | 0.31 | 0.3 | 0.34 | 0.44 |
| Central Tibetan Plateau | 0.15 | 0.25 | 0.11 | 0.05 | -0.19 | -0.09 | -0.10 | 0.14 | -0.12 |
| North China Plain | 0.18 | 0.13 | 0.25 | 0.35 | 0.33 | 0.30 | 0.27 | 0.36 | 0.30 |

[a] Please refer to Fig. 1 for the locations of these regions.

**Table S4.** Comparisons of the RF and RF-STK models in predicting daily ground-level CO concentrations across China during 2013-2016 based on the 10-fold cross-validation.

| Metric[a] | $RF_r$[b] | $RF$[b] | $RF_{rw}$[b] | $RF_w$[b] | $RF_w$-STK[b] | $RF_{rw}$-STK[bc] | $RF_{rw}$-STK[b] |
|---|---|---|---|---|---|---|---|
| $R^2$ | 0.56 | 0.53 | 0.54 | 0.53 | 0.49 | 0.49 | 0.51 |
| Slope | 0.60 | 0.55 | 0.57 | 0.55 | 0.63 | 0.60 | 0.64 |
| RMSE | 0.50 | 0.52 | 0.51 | 0.52 | 0.55 | 0.57 | 0.54 |
| RPE | 46.1% | 48.0% | 47.1% | 48.1% | 51.0% | 53.3% | 50.4% |
| MFB | 0.0832 | -0.013 | -0.0076 | -0.0128 | -0.030 | 0.064 | -0.022 |
| MFE | 0.31 | 0.31 | 0.31 | 0.31 | 0.36 | 0.37 | 0.35 |
| MNB | 0.90 | 0.64 | 0.66 | 0.64 | 0.68 | 0.74 | 0.70 |
| MNE | 1.09 | 0.90 | 0.91 | 0.90 | 0.97 | 1.0 | 0.98 |

[a] $R^2$: coefficient of determination; RMSE: root mean square error (mg m$^{-3}$); RPE: relative prediction error; MFB: mean fractional bias; MFE: mean fractional error; MNB: mean normalized bias; MNE: mean normalized error.

[b] RF: random forest; STK: spatiotemporal kriging. Subscript r indicates a reduced model through variable selection, and subscript w means that the training samples were inversely weighted by the associated population densities. The CO concentrations were log-transformed to train all the models except for $RF_r$ which was trained with the CO concentrations at native scale.

[c] This RF-STK model was developed with the a priori information rather than the MOPITT retrievals.

**Table S5.** Previous studies modeling surface CO concentrations.

| Reference | Model[a] | Study Area | Study Period | Evaluation Metric[b] |
|---|---|---|---|---|
| (Hooghiemstra et al., 2012) | 4D-Var system | Bahia, Brazil | 2007-2009 | $R$=0.6 (daily; prior)
$R$=0.8 (daily; posterior) |
| (Yeganeh et al., 2012) | SVM; PLS-SVM | Tehran, Iran | 2007.01-2011.01 | $R^2$=0.56 (daily; SVM)
$R^2$=0.65 (daily; PLS-SVM) |
| (Hu et al., 2016) | CMAQ | China | 2013.03-2013.12 | MNE=0.59~0.66 (daily)
MFE=0.86~1.02 (daily) |

[a] 4D-Var system: Four-dimensional variational data assimilation system; SVM: support vector machine; PLS-SVM: hybrid model of partial least square and support vector machine; CMAQ: Community Multiscale Air Quality model.

[b] All these studies conducted validation at daily level. Both prior and posterior estimates of the model were evaluated with an independent dataset (Hooghiemstra et al., 2012). Goodness of fit was evaluated in (Yeganeh et al., 2012), and an independent dataset was used for validation in (Hu et al., 2016).

[Figure]

**Figure S1:** Average diurnal pattern in CO concentrations across 1656 monitoring sites for China during 2013-2016. The peak and the valley appeared at 9am and 4pm (Beijing Standard Time). The shaded area represents the standard deviations.

[Figure]

**Figure S2:** Seasonal averages of the ground-level CO concentrations (mg m$^{-3}$) for the whole China, the North China Plain (NCP), the Central Tibetan Plateau (CTP), Naqu, and Qamdo during 2013-2016 based on (a) the MOPITT retrieved surface CO and (b) the observations from the monitoring network. The error bars represent the standard deviations. Naqu and Qamdo are two main cities in CTP (Fig. 8).

[Figure]

**Figure S3:** Coverage rates (%) of MOPITT-CO retrievals for (a) spring, (b) summer, (c) fall, and (d) winter during 2013-2016 across China. The coverage rate at each grid cell was calculated as the percentage of days with MOPITT-CO retrievals in each season.

[Figure]

**Figure S4:** Seasonal means of the averaging-kernel row-sum values associated with the MOPITT retrieved surface CO for (a) spring, (b) summer, (c) fall, and (d) winter during 2013-2016 across China. Small row-sum values indicate strong dependence of the MOPITT retrievals on the a priori information. Please refer to "S.2 Averaging kernel" for more explanation.

[Figure]

**Figure S5:** Annual averages of the MOPITT retrieved surface CO concentrations for (a) 2013, (b) 2014, (c) 2015, and (d) 2016.

[Figure]

**Figure S6:** Annual total CO emissions (t) in (a) 2013, (b) 2014, (c) 2015, and (d) 2016 from anthropogenic sources across China. Due to the data availability, the CO emissions for 2013 and 2015 were linearly interpolated from the available data for 2012, 2014, and 2016 (Li et al., 2017).

[Figure]

**Figure S7:** Evolution of the cross-validation RMSE (mg m⁻³) and $R^2$ for the random forest submodels through the stepwise backward variable selection process.

[Figure]

**Figure S8:** Annual average ground-level CO concentrations from 2013 to 2016 predicted by the $RF_r$, RF, $RF_w$ and $RF_{rw}$-STK models with the MOPITT retrievals. RF: random forest; STK: spatiotemporal kriging. Subscript r indicates a reduced model through variable selection, and subscript w means that the training samples were inversely weighted by the associated population densities. The CO concentrations were log-transformed to train all the models except for $RF_r$ which was trained with the CO concentrations at native scale. The predictions for 2013-2016 by each model are presented from top to bottom rows.

[Figure]

**Figure S9:** Performance of the RF-STK model in predicting daily CO concentrations by regions, years, and seasons. The mean and standard deviation of the root mean square error (RMSE, mg m$^{-3}$) over all the 10-fold cross-validations are presented. The numbers of monitoring sites in Central, East, North, Northeast, Northwest, South, and Southwest China are 267, 255, 307, 171, 159, 278, and 219, respectively. The numbers of monitoring sites in 2013, 2014, 2015, and 2016 are 743, 1041, 1542, and 1603, respectively.

[Figure]

**Figure S10:** Partial dependence plots of the random forest submodel for delineating the relationship between each predictor variable and the ground-level CO concentrations. Partial dependence (Y axis) is the effect of a predictor variable (X axis) on the CO concentrations when the values of all the other predictor variables are fixed at their averages (Friedman, 2001). The subplots are arranged in the order of variable importance. Please refer to Table S1 for the descriptions and units of the predictor variables. The rug plot indicates the data density. Note that the partial dependence estimations are of high uncertainty given low data densities.

[Figure]

**Figure S11:** Coal consumption amounts and energy conversion rates in the sector of power generation and heating for China during 2013-2016 (CSY, 2018).

[Figure]

**Figure S12:** Correlations among the predictor variables and the ground-level CO concentrations, which were measured by the Spearman's rank correlation coefficients. Please refer to Table S1 for the detailed descriptions of the variables.

[Figure]

**Figure S13:** Anthropogenic emission sources of (a) CO, (b) organic carbon (OC), (c) black carbon (BC), (d) volatile organic compound (VOC), (e) NH$_3$, and (f) SO$_2$ for China during 2013-2016 (Li et al., 2017).

[Figure]

**Figure S14:** Total CO emissions (million t) from (a) industry, (b) power, (c) residential, and (d) transportation sectors in each season over China during 2013-2016 (Li et al., 2017).

[Figure]

**Figure S15:** Temporal trends of the population-weighted average ground-level CO concentrations (mg m[-3]) for China during 2013-2016 based on the actual MOPITT retrieved surface CO (blue solid line), the MOPITT a priori surface CO (purple solid line), the predictions made by the RF-STK model using the actual MOPITT retrieved surface CO (red solid line), and the predictions made by the RF-STK model using the MOPITT a priori surface CO (black solid line). The points in different colors represent the deseasonalized monthly averages for deriving the corresponding trend lines. The 95% confidence intervals of the trends are in parentheses followed by the *P* values.

[Figure]

**Figure S16:** Temporal trends of the population-weighted average ground-level CO concentrations (mg m$^{-3}$) for China during 2013-2016 based on the actual (blue solid line) and the bias-adjusted (green solid line) MOPITT retrieved surface CO, as well as the predictions made by the RF-STK model using the actual MOPITT retrieved surface CO (red solid line). The bias correction was carried out according to the mean bias drift of -0.69% per year reported in the previous study (Deeter et al., 2017). The points in different colors represent the deseasonalized monthly averages for deriving the corresponding trend lines. The 95% confidence intervals of the trends (mg m$^{-3}$ per year) are in parentheses followed by the *P* values.

---

## Author Comment (AC2) · 31 Jul 2019

**Response to Anonymous Referee #2:**

It is important to understand the recent trends of ground-level CO concentrations over China for the accurate prediction of air quality. In this study, ground-based and satellite measurements of CO and a state-of-the-art modeling tool have been used to understand the recent trends of CO. I think this study has a high potential of becoming a reference for broad scientific community and for policy makers. However, I was somewhat overwhelmed by the amount of information presented throughout the manuscript with too little explanation. I think there is still room for improvement and below are the suggestions for the authors may want to take into consideration.

**Response:** We highly appreciate this referee's comments. On the basis of the extensive monitoring network and satellite retrievals, we estimated the daily ground-level CO concentrations over China in the recent years by using a refined machine learning model (i.e., RF-STK). To the authors' knowledge, no similar study has been published before. We have thoroughly revised the manuscript by adding more explanations for clarity. Please see the point-by-point responses below. Please note that the all numbers of pages and lines in the responses refer to the revised manuscript. The revised contents are also presented at the end of each response in quotes.

**General Comments**

1.  More background and historical context will be helpful in introduction. What is overall CO trend in China? How were the ground-level CO concentrations measured and used over China in the past? Is the data available in public? What are the short-comings of the in-situ measurements, satellite measurements and modeling studies? The goal of this study needs to be emphasized in introduction more clearly. Proper citations are needed for all the background information, specifically over China. Have there been similar studies like this?

**Response:** Suggestion is taken. In the revision, more background and historical context have been added to Introduction. The detailed responses are listed below.

(a) What is overall CO trend in China?

**Response:** On the basis of the previous studies, the overall ground-level CO trend in China was slowly decreasing in the last decade (Xia et al., 2016; Zheng et al., 2018). Note that the results of these studies were not extensively validated against the observations from the monitoring network. In the revision, we have added the description of the overall CO trend in China to Introduction. Please see Page 2, Lines 19-22.

"In spite of the slow decrease in CO concentrations in recent years based on satellite retrievals (Xia et al., 2016; Zheng et al., 2018), China is still one of the countries with the most severe CO pollution in the world, and the combustion of fossil fuels is the dominant source of anthropogenic CO emissions (Wang et al., 2004; Duncan et al., 2007a)."

(b) How were the ground-level CO concentrations measured and used over China in the past?
**Response:** We have found no literatures about the ground-level CO measurements over China before 2013. Since 2013, the national air quality monitoring network for China has been

measuring the ground-level CO concentrations by using the non-dispersive infrared absorption method and the gas filter correlation infrared absorption method for air quality management. In the revision, we have added the description of the ground-level CO measurement to Introduction. Please see Page 2, Lines 25-28.

"The national air pollution monitoring network in mainland China has been regularly observing ground-level CO concentrations since 2013 (MEPC, 2017) by the non-dispersive infrared absorption method and the gas filter correlation infrared absorption method (CNEMC, 2013), but these site-based measurements are inadequate to represent the spatially continuous distributions of CO (Xu et al., 2014)."

(c) Is the data available in public?
**Response:** The data are publicly available. Please see the revision in Page 3, Lines 33-35.

"We refined the RF-STK model to simulate the daily gridded CO concentrations (0.1° grid with 98341 cells) based on the publicly available datasets, including the ground-level CO monitoring data, the MOPITT retrieved surface CO (MOPITT-CO), and the extensive geographic factors."

(d) What are the short-comings of the in-situ measurements, satellite measurements and modeling studies?
**Response:** The in-situ measurements, which are obtained from the monitoring sites, were of limited spatial coverages. In contrast, the MOPITT satellite measurements have large spatial coverage, but their accuracy tends to be affected by the atmospheric conditions, surface reflectance, and retrieval algorithm. For CTMs, the simulation accuracy is highly associated with the quality of the input data (e.g., emission inventory) and the sophistication of the algorithms. In addition, CTMs are less effective than the RF-STK model in assimilating the measurement data from multiple sources. In the revision, the shortcomings of these three approaches to modeling the CO distributions have been described in introduction. Please see Page 4, Lines 3-6.

"This data assimilation approach compensated the shortcomings of the satellite retrievals (i.e., high uncertainty) and the in-situ measurements (i.e., low spatial coverage) with each other's strengths (i.e., large spatial coverage and high accuracy, respectively), which is more effective and flexible than CTMs in utilizing these measurements."

(e) The goal of this study needs to be emphasized in introduction more clearly.
**Response:** Suggestion is taken. The overall goal of this research is to estimate the spatiotemporal distributions of ground-level CO concentrations across China. To achieve this goal, we refined the hybrid random forest and spatiotemporal kriging (RF-STK) model, which assimilated the data from the national monitoring network and the MOPITT retrievals. We have revised the Introduction by emphasizing the research goal. Please see Page 3, Lines 32-38; Page 4, Lines 1-6.

"The present study aims to estimate the spatiotemporal distributions of ground-level CO concentrations across China during 2013-2016. We refined the RF-STK model to simulate the daily gridded CO concentrations (0.1° grid with 98341 cells) based on the publicly available datasets, including the ground-level CO monitoring data, the MOPITT retrieved surface CO (MOPITT-CO), and the extensive geographic factors. The strategy of inversely weighting the training data by the local population densities was proposed to mitigate the effect of sampling bias towards populous areas for the monitoring network. The spatial resolution of 0.1° has been commonly used for estimating the nationwide distributions of air pollutants in China (Guo et al., 2016; Zhan et al., 2017; Hu et al., 2017b). A machine learning model (i.e., the RF-STK model), for the first time, assimilated the MOPITT-CO with the extensive site-based in-situ CO observations in order to provide more solid information for air quality management. This data assimilation approach compensated the shortcomings of the satellite retrievals (i.e., high uncertainty) and the in-situ measurements (i.e., low spatial coverage) with each other's strengths (i.e., large spatial coverage and high accuracy, respectively), which is more effective and flexible than CTMs in utilizing these measurements. The results of this study are expected to be valuable for air quality management in China."

(f) Proper citations are needed for all the background information, specifically over China.
**Response:** Suggestion is taken. Proper citations have been added to all the background information, specifically over China. Please see Page 2, Lines 25-28.

"The national air pollution monitoring network in mainland China has been regularly observing ground-level CO concentrations since 2013 (MEPC, 2017) by the non-dispersive infrared absorption method and the gas filter correlation infrared absorption method (CNEMC, 2013), but these site-based measurements are inadequate to represent the spatially continuous distributions of CO (Xu et al., 2014)."

(g) Have there been similar studies like this?
**Response:** On the basis of our extensive literature review, the previous studies used either satellite retrievals or chemical transport models to investigate the spatiotemporal distributions of CO in China (Peng et al., 2007; Zhao and Chi, 2018). Unlike the previous studies, we assimilated the MOPITT data and the observations from the monitoring network in order to derive the daily ground-level CO concentrations across China, exhibiting highlighted predictive performance. In the revision, we have emphasized the significance of this approach to estimating the spatiotemporal distribution of ground-level CO. Please see Page 4, Lines 1-3.

"A machine learning model (i.e., the RF-STK model), for the first time, assimilated the MOPITT-CO with the extensive site-based in-situ CO observations in order to provide more solid information for air quality management."

References for this response

CNEMC: Technical Specifications for Installation and Acceptance of Ambient air Quality Continuous Automated Monitoring System for SO2, NO2, O3 and CO, Ministry of Ecology and Environment of the People's Republic of China, 2013.

Guo, Y., Zeng, H., Zheng, R., Li, S., Barnett, A. G., Zhang, S., Zou, X., Huxley, R., Chen, W., and Williams, G.: The association between lung cancer incidence and ambient air pollution in China: A spatiotemporal analysis, Environ. Res., 144, 60-65, 10.1016/j.envres.2015.11.004, 2016.

Hu, X., Belle, J. H., Meng, X., Wildani, A., Waller, L. A., Strickland, M. J., and Liu, Y.: Estimating PM2.5 Concentrations in the Conterminous United States Using the Random Forest Approach, Environ. Sci. Technol., 51, 6936-6944, 10.1021/acs.est.7b01210, 2017b.

Peng, L., Zhao, C., Lin, Y., Zheng, X., Tie, X., and Chan, L. Y.: Analysis of carbon monoxide budget in North China, Chemosphere., 66, 1383-1389, 10.1016/j.chemosphere.2006.09.055, 2007.

Xia, Y., Zhao, Y., and Nielsen, C. P.: Benefits of China's efforts in gaseous pollutant control indicated by the bottom-up emissions and satellite observations 2000–2014, Atmos. Environ., 136, 43-53, 10.1016/j.atmosenv.2016.04.013, 2016.

Zhao, X., and Chi, T.: Spatial Distribution and Temporal Variation of Tropospheric CO Concentration over China Based on MOPITT Measurements (in Chinese with English abstract), Environ. Sci. Technol., 41, 71-76, 10.19672/j.cnki.1003-6504.2018.04.013, 2018. Tang, W., Arellano, A. F., Gaubert, B., Miyazaki, K., and Worden, H. M.: Satellite data reveal a common combustion emission pathway for major cities in China, Atmos. Chem. Phys., 19, 4269-4288, 10.5194/acp-19-4269-2019, 2019.

Zheng, B., Chevallier, F., Ciais, P., Yin, Y., Deeter, M. N., Worden, H. M., Wang, Y., Zhang, Q., and He, K.: Rapid decline in carbon monoxide emissions and export from East Asia between years 2005 and 2016, Environ. Res. Lett., 13, 044007, 10.1088/1748-9326/aab2b3, 2018.

2. Why was the 2013-2016 period chosen? Is this period long enough to provide us reliable trend analysis? Wasn't the air pollution over China more severe before 2013?

**Response:** We chose the period of 2013-2016 due to the data availability. The national air quality monitoring network has been monitoring the ground-level CO and other air pollutants such as PM2.5 over China since 2013. While the air pollution in China was considered to be more severe in earlier year (Krotkov et al., 2016), no large-scale monitoring data were available before 2013 to train the RF-STK model. While the period of 2013-2016 might not be sufficient for analyzing the long-term trend, the nationwide data for this period provide valuable information on the spatiotemporal distributions of CO concentrations across China. In the revision, we have added explanation of choosing 2013-2016 as the study period and discussed long-term analyses for future work. Please see Page 12, Line 14-18.

"We chose the period of 2013-2016 for this study due to the data availability. While the air pollution in China was severer in earlier years (Krotkov et al., 2016), no large-scale monitoring data were available before 2013 for training the RF-STK model. Back-extrapolation such as a previous study (Gulliver et al., 2016) may be conducted based on MOPITT-CO since 2000, whereas the issue of bias drift is currently difficult to deal with."

References for this response

Gulliver, J., de Hoogh, K., Hoek, G., Vienneau, D., Fecht, D., and Hansell, A.: Back-extrapolated and year-specific NO2 land use regression models for Great Britain - Do they yield different exposure assessment?, Environ. Int., 92-93, 202-209, 10.1016/j.envint.2016.03.037, 2016.

Krotkov, N. A., McLinden, C. A., Li, C., Lamsal, L. N., Celarier, E. A., Marchenko, S. V., Swartz, W. H., Bucsela, E. J., Joiner, J., Duncan, B. N., Boersma, K. F., Veefkind, J. P., Levelt, P. F., Fioletov, V. E., Dickerson, R. R., He, H., Lu, Z., and Streets, D. G.: Aura OMI observations of regional SO 2 and NO 2 pollution changes from 2005 to 2015, Atmos. Chem. Phys., 16, 4605-4629, 10.5194/acp-16-4605-2016, 2016.

3.  Since this study is focused on a smaller region, the importance of higher spatial and temporal resolution measurements and also higher resolution model should be mentioned as well. Considering MOPITT's large footprint (22x22 $km^2$), is MOPITT the best fit for this type of study?

**Response:** Measurements/models with high spatial and temporal resolutions are important to studies focusing on small regions. This study aims to estimate the spatiotemporal distributions of daily ground-level CO concentrations across the whole China during 2013-2016, with detailed analyses on smaller regions like the Central Tibetan Plateau (CTP). On the basis of the comprehensive literature review, MOPITT-CO is the best publicly available satellite data for modelling the CO distribution on a national scale. For the smaller regions, MOPITT's spatial resolution (22 km at nadir) is relatively coarse, and the TROPOspheric Monitoring Instrument (TROPOMI) onboard the Sentinel-5P satellite with a spatial resolution of 7km×3.5km better fits the need. Nevertheless, TROPOMI have been providing the CO column density data since July, 2018, which are exclusive of the period 2013-2016 for the present study. In the revision, we have discussed the importance of measurements and models with higher spatial and temporal resolution to small-scale analyses. We have also justified the choice of MOPITT for this study. Please see Page 12, Lines 18-24.

"In addition, measurements or model predictions with high spatial (e.g., 1 km) and temporal resolutions (e.g., 1 hour) are important to studies focusing on small regions, such as CTP in this study. In spite of its relative coarse resolution (22 km at nadir), the MOPITT product provided the best publicly available satellite-based measurements of surface CO for China during 2013-2016. Since July of 2018, the TROPOspheric Monitoring Instrument onboard the Sentinel-5P satellite has been providing the CO product at a higher resolution of 7 km × 3.5 km (Borsdorff et al., 2018), which may replace MOPITT-CO in the RF-STK model in order to make predictions at a higher resolution."

Reference for this response

Borsdorff, T., Aan de Brugh, J., Hu, H., Aben, I., Hasekamp, O., and Landgraf, J.: Measuring Carbon Monoxide With TROPOMI: First Results and a Comparison With ECMWF-IFS Analysis Data, Geophys. Res. Lett., 45, 2826-2832, 10.1002/2018gl077045, 2018.

4. I think there are many nice figures and tables included in the manuscript and supplement material. I would recommend reorganizing the figures with consistency. I find myself going back and forth the manuscript and supplement material trying to find the figures. I would also recommend to spend more time on describing figures and tables. Each figure contains more information than just being cited in the parenthesis.

**Response:** Suggestion is taken. In the revision, the orders of the figures have been rearranged in order to be consistent with the orders of their first occurrences in the manuscript. More explanations have been to the text and the figure captions. Please see Page 10, Lines 7-16 for examples.

"As a machine learning approach, the RF-STK model exhibited stable performance across regions and seasons (Fig. S9), which was comparable or superior to the previous CTMs or statistical methods simulating ground-level CO concentrations (Table S5). As the simulation areas and episodes were considerably different among these studies, their predictive performance was not strictly comparable. A hybrid statistical model (partial least square and support vector machine) exhibited decent goodness-of-fit in simulating daily CO concentrations in Tehran, Iran, with fitting $R^2$=0.65 (Yeganeh et al., 2012). For the CTM study in Bahia, Brazil, the accuracy of the posterior estimation improved largely after incorporating the surface observations into the priori state (Hooghiemstra et al., 2012). In the absence of nationwide statistical modeling work, only CTM studies were found for modeling CO at large scale in China. A previous CTM work for China underestimated the ground-level CO concentrations by 67.2% on average (Hu et al., 2016), which might be due to the underestimation of CO emissions."

"**Figure S10:** Partial dependence plots of the random forest submodel for delineating the relationship between each predictor variable and the ground-level CO concentrations. Partial dependence (Y axis) is the effect of a predictor variable (X axis) on the CO concentrations when the values of all the other predictor variables are fixed at their averages (Friedman, 2001). The subplots are arranged in the order of variable importance. Please refer to Table S1 for the descriptions and units of the predictor variables. The rug plot indicates the data density. Note that the partial dependence estimations are of high uncertainty given low data densities."

5. The results are presented here in the form of numbers and tables, which might give a quantitative information. However, it is somewhat challenging to see what the scientific messages are. I would recommend including tables only when is absolutely necessary. Figures are easier to understand otherwise.

**Response:** Suggestion is taken. In the revision, we have replaced Tables 2, S1, S3, and S9 by figures for addressing the messages more intuitively. For the remaining tables, we think it is necessary to show the numbers in the form of tables. Please see Figures 2, S2, and S13 for the revisions.

[Figure]

**Figure 2:** (a) Seasonal and (b) annual means of the population-weighted average ground-level CO concentrations (mg m$^{-3}$) during 2013-2016 for China predicted by the RF-STK model. The error bars (standard deviations) stand for the spatial variations.

[Figure]

**Figure S2:** Seasonal averages of the ground-level CO concentrations (mg m⁻³) for the whole China, the North China Plain (NCP), the Central Tibetan Plateau (CTP), Naqu, and Qamdo during 2013-2016 based on (a) the MOPITT retrieved surface CO and (b) the observations from the monitoring network. The error bars represent the standard deviations. Naqu and Qamdo are two main cities in CTP (Fig. 8).

[Figure]

**Figure S13:** Anthropogenic emission sources of (a) CO, (b) organic carbon (OC), (c) black carbon (BC), (d) volatile organic compound (VOC), (e) NH3, and (f) SO2 for China during 2013-2016 (Li et al., 2017).

6. Section 3.3 and 3.4 contain mainly technical information and the figure numbers do not seem to have any particular order. Rewriting those sections with more explanation will help.

**Response:** Suggestion is taken. In the revision, we have revised Sections 3.3 and 3.4 for clarity. The figure numbers have been corrected to the order of their first occurrences in the text. Please see Page 9, Lines 13-16; Page 9, Lines 22-24; Page 9, Lines 37-38; Page 10, Lines 1-6; Page 10, Lines 7-16; Page 10, Lines 18-27; Page 10, Lines 34-38; Page 11, Lines 1-5.

Page 9, Lines 13-16.
"Through the variable selection, a concise structure of the RF submodel was achieved, and the spurious prediction details (e.g., the sharp boundaries) were mitigated (Fig. S7). For instance, the RF submodel with all the predictors generated sharp boundaries circling the desert areas in Northwest China, which became blurred in the predictions made by the reduced RF submodel with the selected predictors (Fig. S8)."

Page 9, Lines 22-24.
"Compared to the original RF-STK model proposed in the previous study (Zhan et al., 2018), this refined RF-STK model had two major modifications, including sample weighting and logarithm transformation of the response variable (i.e., ground-level CO observations in the present study)."

Page 9, Lines 37-38; Page 10, Lines 1-6.

[revised manuscript text omitted]

**Specific Comments**
7. P2, L12 It is not clear the meaning of 'overlooked' here.
**Response:** The word 'overlooked' here means that the high concentrations of ground-level CO in the Central Tibetan Plateau were not detected by the MOPITT retrievals. In the revision, we have revised this sentence for clarity. Please see Page 2, Lines 11-12.

"The hotspots in the Central Tibetan Plateau where the CO concentrations were underestimated by MOPITT-CO were apparent in the RF-STK predictions."

8. P2, L25 A reference or more information needed.
**Response:** Suggestion is taken. While most of the monitoring sites were located in urban areas of major cities, the vast rural and suburban region of China was not covered by the air pollution monitoring network. Considering the spatial heterogeneity of CO concentrations, the site-based measurements alone were inadequate to represent the spatial distributions of ground-level CO concentrations. More information with a reference has been added to the revised manuscript. Please see Page 2, Lines 25-28.

"The national air pollution monitoring network in mainland China has been regularly observing ground-level CO concentrations since 2013 (MEPC, 2017) by the non-dispersive infrared absorption method and the gas filter correlation infrared absorption method (CNEMC, 2013), but these site-based measurements are inadequate to represent the spatially continuous distributions of CO (Xu et al., 2014)."

Reference for this response
CNEMC: Technical Specifications for Installation and Acceptance of Ambient air Quality Continuous Automated Monitoring System for SO2, NO2, O3 and CO, Ministry of Ecology and Environment of the People's Republic of China, 2013.
MEPC Air quality daily report for China: http://datacenter.mep.gov.cn/, access: 30 Jan 2017, 2017.

9. P2, L27 CTM > CTMs

**Response:** Suggestion is taken. The acronym "CTM" has been changed to "CTMs". Please see Page 2, Lines 29-30.

"Chemical Transport Models (CTMs) have been employed to estimate ground-level CO concentrations (Arellano and Hess, 2006; Hu et al., 2016)."

10. P2, L30-31 Any references for this?
**Response:** Yes, we have added two references (Li et al., 2010; Hu et al., 2017a) for this statement about the predictive performance of CTMs. Please see Page 3, 32-34.

"The predictive performance of CTMs tends to be affected by uncertainties in the simulation algorithms and the emission inventories (Li et al., 2010; Hu et al., 2017a)."

References for this response
Li, M. J., Chen, D. S., Cheng, S. Y., Wang, F., Li, Y., Zhou, Y., and Lang, J. L.: Optimizing emission inventory for chemical transport models by using genetic algorithm, Atmos. Environ., 44, 3926-3934, 10.1016/j.atmosenv.2010.07.010, 2010.
Hu, J., Li, X., Huang, L., Ying, Q., Zhang, Q., Zhao, B., Wang, S., and Zhang, H.: Ensemble prediction of air quality using the WRF/CMAQ model system for health effect studies in China, Atmos. Chem. Phys., 17, 13103-13118, 10.5194/acp-17-13103-2017, 2017a.

11. P3, L3-18 More current references for all the satellite instruments are need here.
**Response:** Suggestion is taken. More current references for all the satellite instruments have been added, including (Worden et al., 2013a, Worden et al., 2013b; Deeter et al., 2014; Jiang et al., 2015; Strode et al., 2016; Deeter et al., 2017) for MOPITT, (Wang et al., 2018) for AIRS, (Kopacz et al., 2010; Ul-Haq et al., 2016) for SCIAMACHY, (Barret et al., 2016) for IASI. Please see Page 3, Lines 5-20.

[revised manuscript text omitted]

12. P3, L10 Deeter et al. (2014, 2017) should be included here.

**Response:** Suggestion is taken. Deeter et al. (2014, 2017) have been included in the suggested sentence. Please see Page 3, Lines 12-13.

"Among the abovementioned satellite instruments, MOPITT is one of few sensors that are capable of measuring ground-level CO based on the instantaneous multispectral retrievals (Streets et al., 2013; Deeter et al., 2014; Deeter et al., 2017)."

Reference for this response
Deeter, M. N., Martínez-Alonso, S., Edwards, D. P., Emmons, L. K., Gille, J. C., Worden, H. M., Sweeney, C., Pittman, J. V., Daube, B. C., and Wofsy, S. C.: The MOPITT Version 6 product: algorithm enhancements and validation, Atmos. Meas. Tech., 7, 3623-3632, 10.5194/amt-7-3623-2014, 2014.
Deeter, M. N., Edwards, D. P., Francis, G. L., Gille, J. C., Martínez-Alonso, S., Worden, H. M., and Sweeney, C.: A climate-scale satellite record for carbon monoxide: the MOPITT Version 7 product, Atmos. Meas. Tech., 10, 2533-2555, 10.5194/amt-10-2533-2017, 2017.

13. P6, L18-19 I wonder what is causing the sparse coverage over China. Also, how much coverage is considered to be enough or limiting here?

**Response:** In addition to the reflectance condition and satellite orbit, the narrower swath width of MOPITT (640 km) compared to MODIS (2330 km) is a main factor causing the sparse coverage. While MOPITT and MODIS are both onboard the Terra satellite, the repeat cycle of MOPITT is approximately 3 days compared to 1-2 days of MODIS (Edwards et al., 2004). Since this study aims to derive the daily ground-level CO concentrations, the ideal repeat cycle would be 1 day or shorter. In the revision, we have added more discussion on the issue of sparse coverage. Please see Page 7, Lines 35-36; Page 8, Lines 1-4.

"In addition to the reflectance condition and the satellite orbit, the narrower swath width of MOPITT (640 km) compared to the Moderate Resolution Imaging Spectroradiometer (MODIS) with a swath width of 2330 km was one of the main factors causing the sparse coverage. While MOPITT and MODIS are both onboard the Terra satellite, the measurement repeat cycle of MOPITT is approximately 3 days compared to 1-2 days of MODIS (Edwards et al., 2004). The sparse coverages of MOPITT-CO limit its utility for representing time-series of daily CO concentrations across China."

Reference for this response
Edwards, D. P., Emmons, L. K., Hauglustaine, D. A., Chu, D. A., Gille, J. C., Kaufman, Y. J., Pétron, G., Yurganov, L. N., Giglio, L., Deeter, M. N., Yudin, V., Ziskin, D. C., Warner, J., Lamarque, J. F., Francis, G. L., Ho, S. P., Mao, D., Chen, J., Grechko, E. I., and Drummond, J. R.: Observations of carbon monoxide and aerosols from the Terra satellite: Northern Hemisphere variability, J. Geophys. Res-Atmos., 109, 10.1029/2004JD004727, 2004.

14. P6, L26-30 For clarity, the authors need to include citations or data sources here. Also, how does the spatial coverage affect the bias and uncertainties?

**Response:** These descriptive statistics of the ground-level CO observations are summarized from the monitoring data obtained from the National Air Quality Monitoring Network for mainland China (MEPC, 2017), the Environmental Protection Department of Hong Kong (EPDHK, 2017), and the Environmental Protection Administration of Taiwan (EPAROC, 2017). The data sources have been added to the revised manuscript for clarity. In addition, the effects of the spatial coverage of monitoring stations on the statistics of CO concentrations have been explained in the revision. The air quality monitoring network is of considerable sampling bias, i.e., most of the stations are located in major cities of eastern China, which would introduce bias to the estimates of national or regional average concentrations. For instance, the national average concentration would be overestimated by simply calculating the averages of all the monitoring data, since the concentrations in remote areas tend to be lower. Please see Page 7, Lines 23-30 for the revision.

"Note that the scale of monitoring network was not constant, and the number of monitoring sites grew from 743 to 1603 during these four years (MEPC, 2017; EPAROC, 2017; EPDHK, 2017). However, the monitoring stations were still sparse in the western China throughout the monitoring period, and most of the stations were located in the major cities of the eastern China (Fig. 1). The spatially imbalanced monitoring (i.e., sampling bias) therefore tends to introduce bias to the spatiotemporal statistics of CO concentrations (Boria et al., 2014). For instance, the national average concentrations would be overestimated if they were simply determined as the averages of all the monitoring data, as the CO concentrations were generally lower in remote areas."

15. P7, L4 Why is the correlation coefficient higher in winter? What does the seasonal dependency in the correlation coefficients mean?

**Response:** The stronger correlation between the ground-level CO and MOPITT-CO in winter is mainly attributed to the higher CO concentrations in winter than the other seasons (Table S3). Higher concentrations lead to higher signal-to-noise ratios, which is more favorable for MOPITT to capture the spatiotemporal variation in the ground-level CO concentrations. The seasonal dependency in the correlation coefficients means that the MOPITT retrievals are more sensitive to high CO concentrations. We have added the explanation to the revised manuscript. Please see Page 8, Lines 22-24.

"The stronger correlation in winter was mainly attributed to the higher signal-to-noise ratios accompanied with the higher CO concentrations, reflecting that the MOPITT-CO was more sensitive in measuring high CO concentrations."

16. P7, L7-11 For the MOPITT retrievals over the Tibetan Plateau, the authors might want to contact the MOPITT science team and seek for advice. Including the latest development in their retrieval methods will be useful here.

**Response:** Thanks for the suggestion. In this study, we compared the MOPITT retrievals with the ground-based observations from the monitoring network, with more focused analyses on the Central Tibetan Plateau (CTP). In the future, we will try to contact the MOPITT science team and seek for their advice on explaining/resolving the discrepancy between the MOPITT retrievals and the ground-based observations.

17. P7, L39 Table S6 has so much information and it is not explained in the text at all.
**Response:** The entries of Table S5 in the Supplementary Data have been refined to three studies specifically modeling ground-level CO concentrations. Two studies used Chemical Transport Models (CTMs), and one study used a statistical model. As the modeling zones and episodes were different among these studies, their predictive performance was not strictly comparable. The purpose of presenting Table S5 was to give a brief summary of the modeling methods and performance for surface CO concentrations. In the revision, we have explained Table S5 in more detail. Please see Page 10, Lines 7-16.

"As a machine learning approach, the RF-STK model exhibited stable performance across regions and seasons (Fig. S9), which was comparable or superior to the previous CTMs or statistical methods simulating ground-level CO concentrations (Table S5). As the simulation areas and episodes were considerably different among these studies, their predictive performance was not strictly comparable. A hybrid statistical model (partial least square and support vector machine) exhibited decent goodness-of-fit in simulating daily CO concentrations in Tehran, Iran, with fitting $R^2$=0.65 (Yeganeh et al., 2012). For the CTM study in Bahia, Brazil, the accuracy of the posterior estimation improved largely after incorporating the surface observations into the priori state (Hooghiemstra et al., 2012). In the absence of nationwide statistical modeling work, only CTM studies were found for modeling CO at large scale in China. A previous CTM work for China underestimated the ground-level CO concentrations by 67.2% on average (Hu et al., 2016), which might be due to the underestimation of CO emissions."

18. P10, L1 Explain what 'importance of coal consumption' specifically means and how is related to CO trend. Do people use less coal than before? Is combustion efficiency improving? Does this have a seasonal dependency?
**Response:** The phrase of "importance of coal consumption" means that the coal consumption could be one of the major sources of atmospheric CO in China. Coal consumption accounted for approximately 70% of the total energy use in China during 2013-2016 (CSY, 2018). As the major energy consumers, the industrial and residential sectors emitted a large quantity of CO, accounting for 41 and 39% of the total anthropogenic CO emissions, respectively (Fig. S13). The CO emissions from these sectors exhibited a strong seasonality, especially for the residential sector (Fig. S14). In winter, more coal was consumed for heating, resulting in higher CO emissions and hence more severe CO pollution (Li et al., 2017). From 2013 to 2016, the energy conversion rate increased, while the amount of coal consumption decreased (Fig. S6). The decrease rate of coal consumption was similar to the decrease rate of ground-level CO concentrations. On the basis of the above results, we inferred that the decrease in coal

consumption played an important role in the mitigation of CO pollution in China. In the revision, we have added more explanations of the "importance of coal consumption". Please see Page 11, Lines 21-27; Figures S6, S11 and S14 in the Supplementary Data.

"The relative decrease rate of 4.4% was similar to the 3.8% drop of coal consumption for China during 2013-2016, suggesting the potentially important contribution of decrease in coal consumption (partially due to improved energy conversion efficiency; Fig. S11) to the mitigation of CO pollution (CSY, 2018). Coal consumption accounted for approximately 70% of the total energy use in China. As the major energy consumers, the industrial and residential sectors contributed 41 and 39% of the total anthropogenic CO emissions, respectively (Fig. S13). More coal was consumed for residential heating in winter, causing higher CO emissions and more severe air pollution (Fig. S14)."

[Figure]

**Figure S6:** Annual total CO emissions (t) in (a) 2013, (b) 2014, (c) 2015, and (d) 2016 from anthropogenic sources across China. Due to the data availability, the CO emissions for 2013 and 2015 were linearly interpolated from the available data for 2012, 2014, and 2016 (Li et al., 2017).

[Figure]

**Figure S11:** Coal consumption amounts and energy conversion rates in the sector of power generation and heating for China during 2013-2016 (CSY, 2018).

[Figure]

**Figure S14:** Total CO emissions (million t) from (a) industry, (b) power, (c) residential, and (d) transportation sectors in each season over China during 2013-2016 (Li et al., 2017).

19.  P10, L6-7 Why are the trends estimated by MOPITT lower?

**Response:** As the trend underestimation was seldomly reported or discussed in the literatures, we proposed an explanation based on our results. The a priori information which was the same across the years might contribute to the trend underestimation by MOPITT. The underestimation was more apparent for the regions with weaker average kernels (Figs. 9 and

S4), suggesting that higher dependence on the a priori led to more severe underestimation of the trends. In the revision, we have added the explanation to the trend underestimation by MOPITT. Please see Page 11, Lines 37-38; Page 12, Lines 1-3.

"The trend underestimation by MOPITT-CO might be largely due to the setting that the a priori information was the same across the years (Dekker et al., 2017). We found that the trend underestimation tended to be more severe for the regions with weaker averaging kernels (Figs. 9 and S5), which was analogous to the phenomenon that the predictions made by the RF-STK model with the a priori information exhibited a slower decreasing rate (-2.06% per year) than the model with MOPITT-CO (Fig. S15)."

20. P10, L10 'The refined RF-STK predictions that assimilates the MOPITT-CO with ground-level CO observations provide more solid information for decision making.' I think this sentence is very important and should be in introduction.

**Response:** Suggestion is taken. We have moved this sentence to the last paragraph of introduction. Please see Page 4, Lines 1-3.

"A machine learning model (i.e., the RF-STK model), for the first time, assimilated the MOPITT-CO with the extensive site-based in-situ CO observations in order to provide more solid information for air quality management."

21. P10, L29 'such as refining the prior status assigned to the overlooked hotspots in the Central Tibetan Plateau.'- I wonder how the results in this study can be utilized in improving MOPITT retrievals?

**Response:** The predicted spatiotemporal distribution of ground-level CO concentrations may be used to refine the emission inventory of CO for the Central Tibetan Plateau (CTP) so that the a priori of MOPITT can be improved. Given the strong dependence on the a priori for CTP (please see the response to comment #2 of anonymous referee #1), the MOPITT ground-level retrievals would be consequently improved. In the revision, we have added explanation of utilizing the results in this study for improving the MOPITT retrievals. Please see Page 13, Lines 4-6.

"The present study provides important information for improving the accuracy of MOPITT retrievals, such as refining the a priori assigned to the CO hotspots in CTP constrained by the RF-STK predictions."

**Figures and Tables**
22. I am wondering how the figures in the manuscript and supplement material are divided. It seems like the figure descriptions in the text has no particular order.

**Response:** The figures in the manuscript and supplement are divided based on their relative importance to the main contents of this study. For instance, as the Central Tibetan Plateau (CTP) is a focused region in this study, the enlarged map of CTP is shown in the manuscript. In

addition, the order of the figures has been corrected according to the orders of their first occurrences in the manuscript.

23. Figure 1 What is Heihe-Tengchong line? And what is the purpose of showing here? I do not see any relevance of inserting the South China Sea map as we are only considering the ground measurements stations here. I recommend removing the inserted map.

**Response:** The Heihe-Tengchong line is a "geo-demographic demarcation line" imagined by a Chinese population geographer, reflecting the disparity in the population distribution. Around 95% of the population live to the east of the line. The purpose of showing the line here is to highlight the imbalanced distribution of the monitoring sites associated with the population distribution. This map is produced from a commonly used template with the South China Sea placed at the bottom right corner. We would like to keep the integrity of this map template. Thanks for your understanding. In the revision, we have added the explanation of the Heihe-Tengchong line to the caption of Figure 1. Please see Figure 1 on Page 22.

"The red dashed line represents the Heihe-Tengchong Line, which is an imagined "geo-demographic demarcation line" reflecting the disparity in the population distribution. Around 95% of the population live to the east of the line, where 82% of the monitoring sites are located."

24. Table 1 Higher correlation coefficients (> 0.9) can be marked as bold or shaded numbers for better visibility.

**Response:** Suggestion is taken. We have marked higher correlation coefficients (> 0.9) as bold for better visibility. Please see Table 1 on Page 20.

25. Table 2 can be replaced by bar-graphs, if it's possible. This applies to other tables included in supplements.

**Response:** Suggestion is taken. In the revision, we have replaced Tables 2, S1, and S3 by bar-graphs. Table S9 has been replaced by a pie chart. Please see Page 20; Pages 9 and 20 in the Supplementary Data.

[Figure]

**Figure 2:** (a) Seasonal and (b) annual means of the population-weighted average ground-level CO concentrations (mg m$^{-3}$) during 2013-2016 for China predicted by the RF-STK model. The error bars (standard deviations) stand for the spatial variations.

[Figure]

**Figure S2:** Seasonal averages of the ground-level CO concentrations (mg m$^{-3}$) for the whole China, the North China Plain (NCP), the Central Tibetan Plateau (CTP), Naqu, and Qamdo during 2013-2016 based on (a) the MOPITT retrieved surface CO and (b) the observations from the monitoring network. The error bars represent the standard deviations. Naqu and Qamdo are two main cities in CTP (Fig. 8).

[Figure]

**Figure S13:** Anthropogenic emission sources of (a) CO, (b) organic carbon (OC), (c) black carbon (BC), (d) volatile organic compound (VOC), (e) NH3, and (f) SO2 for China during 2013-2016 (Li et al., 2017).

26. Figure S1 How is the seasonal coverage calculated?

**Response:** The seasonal coverage of a grid cell is calculated as the percent of days having MOPITT-CO retrievals for that season during 2013-2016. The explanation of the seasonal coverage has been added to the caption of Figure S3 in the Supplementary Data.

"The coverage rate at each grid cell was calculated as the percentage of days with MOPITT-CO retrievals in each season."

27. Figure S2 Standard deviation (uncertainty) can be added here.

**Response:** Suggestion is taken. In the revision, we have revised Figure S1 by adding the standard deviations. Please see Figure S1 in the Supplementary Data.

[Figure]

**Figure S1:** Average diurnal pattern in CO concentrations across 1656 monitoring sites for China during 2013-2016. The peak and the valley appeared at 9am and 4pm (Beijing Standard Time). The shaded area represents the standard deviations.

28. Figure S10 What is partial dependence plot? Also, what are the x and y axes on this plot?
**Response:** Partial dependence plots are derived from the random forest submodel for delineating the relationships between the predictor variable and the response variable (i.e., ground-level CO concentrations in this study). Partial dependence (Y axis) is the effect of a predictor variable (X axis) on the CO concentrations when the values of all other predictor variables are set to their averages. In the revision, we have added more explanations of partial dependence plot to the caption of Figure S10. Please see Figure S10 in the Supplementary Data.

"**Figure S10:** Partial dependence plots of the random forest submodel for delineating the relationship between each predictor variable and the ground-level CO concentrations. Partial dependence (Y axis) is the effect of a predictor variable (X axis) on the CO concentrations when the values of all the other predictor variables are fixed at their averages (Friedman, 2001). The subplots are arranged in the order of variable importance. Please refer to Table S1 for the descriptions and units of the predictor variables. 
[revised manuscript text omitted]

The data of MOPITT retrieved surface CO (MOPITT-CO) are processed with the temporal and spatial convolution to filter noises and fill data gaps. In the first step, the temporal convolution with a 1-dimensioanl Gaussian kernel is employed to process the MOPITT-CO data for each grid cell:

$$M_T(t_0) = \sum_t [M(t) \cdot W_T(t_0 - t)] / \sum_t W_T(t_0 - t) \tag{1}$$

where $M_T(t_0)$ is the output value on day $t_0$ processed by the temporal convolution, $M(t)$ is the original MOPITT-CO value on day $t$, and $W_T(t_0 - t)$ is the weighting factor determined by the 1-dimensional Gaussian function:

$$W_T(t_0 - t) = exp[-(t_0 - t)^2 / (2\sigma_T^2)] \tag{2}$$

where the standard deviation ($\sigma_T$) is set to 60 according to the sensitivity analysis on the completeness and smoothness of the processed data.

In the second step, the spatial convolution with a 2-dimensional Gaussian kernel is employed to process the output from the previous step day by day:

$$M_{TS}(x_0, y_0) = \sum_{x,y} [M_T(x, y) \cdot W_S(x_0 - x, y_0 - y)] / \sum_{x,y} W_S(x_0 - x, y_0 - y) \tag{3}$$

where $M_{TS}(x_0, y_0)$ is the output value for cell $(x_0, y_0)$ processed by the spatial convolution, $M_T(x, y)$ is the processed MOPITT-CO value from the first step for cell $(x, y)$, and $W_S(x_0 - x, y_0 - y)$ is the weighting factor determined by the 2-dimensional Gaussian function:

$$W_S(x_0 - x, y_0 - y) = exp\{-[(x_0 - x)^2 + (y_0 - y)^2)] / 2\sigma_S^2\} \tag{4}$$

where the standard deviation ($\sigma_S$) is set to 0.1 according to the sensitivity analysis on the completeness and smoothness of the processed data.

**S.2 Averaging kernel**

The averaging kernel (matrix $A$) adjusts the weights of the "true" state (vector $x$) and the a priori (vector $x_a$) in deriving the MOPITT CO retrievals (vector $\hat{x}$) (Deeter et al., 2003; Rodgers, 2000).

$$\hat{x} \approx Ax + (I - A)x_a \tag{5}$$

where $I$ is the identity matrix. Each row of $A$ corresponds to a vertical layer of the CO profile, and the sum of a row shows the overall dependence of the MOPITT CO retrieval at that layer on the a priori information. A small row-sum value indicates strong dependence on the a priori information.

**S.3 Algorithm of random forests (Breiman, 2001)**

For *tree* = 1 to *N* (e.g., 500 trees in this study):

- ◆ Randomly draw a sample from the training data with replacement through bootstrapping;
- ◆ A tree is grown from a single node, and the following steps are repeated until the minimum number of observations is present at each terminal node:
  - ✧ Randomly select a subset of predictors to be considered at each split;
  - ✧ Find the split that reduces the squared error the most;

Average the predictions made by all the decision trees as the output of the random forest.

**S.4 Environmental condition data**

◆ The daily weather conditions, including atmospheric pressure, air temperature, precipitation, evaporation, insolation duration, and wind speed, were obtained from 839 meteorological stations (CMA, 2017).

◆ The elevation data were retrieved from the Shuttle Radar Topography Mission (SRTM) database (Jarvis et al., 2016).

◆ The data of population density, road density, and land use were extracted from the Gridded Population of the World, the OpenStreetMap, and the GlobeLand30 databases, respectively (CIESIN, 2016; OSP, 2016; Jun et al., 2014).

◆ The daily planetary boundary height (PBLH) data were obtained from the Modern-Era Retrospective Analysis for Research and Application (GMAO, 2015).

◆ The Normalized Difference Vegetation Index (NDVI) data were retrieved from the Moderate Resolution Imaging Spectroradiometer (MODIS) satellite retrievals (Didan et al., 2015).

◆ The anthropogenic emission inventories were obtained from the Multi-resolution Emission Inventory for China (MEIC) database (Li et al., 2017). Due to the data availability, the emissions for 2013 and 2015 were linearly interpolated from the available emission data for 2012, 2014, and 2016.

**Table S1.** List of variable symbols and definitions.

| Symbol | Unit | Variable definition | Spatial[a] | Temporal[a] | Convolution[b] |
|---|---|---|---|---|---|
| MOPITT | molecule cm$^{-2}$ | MOPITT-retrieved CO surface mixing ratio | 0.25° | Day | Temporal and Spatial |
| DOY | - | Day of year | - | - | - |
| YEAR | - | Year | - | - | - |
| EVP | mm | Evaporation | Point | Day | - |
| PRE | mm | Precipitation | Point | Day | - |
| PRS | hPa | Atmospheric pressure | Point | Day | - |
| RHU | % | Relative humidity | Point | Day | - |
| SSD | hour | Sunshine duration | Point | Day | - |
| TEM | °C | Temperature | Point | Day | - |
| WIN | m s$^{-1}$ | Wind speed | Point | Day | - |
| PBLH | Km | Planetary boundary layer height | 0.625°×0.5° | Day | - |
| ELV | M | Elevation | 90 m | - | Spatial |
| NDVI | - | Normalized Difference Vegetation Index | 250 m | 8 Days | Spatial |
| POP | people km$^{-2}$ | Population density | 30" | - | Spatial |
| LU10 | % | Cultivated land area | 30 m | - | Spatial |
| LU20 | % | Forest area | 30 m | - | Spatial |
| LU30 | % | Grassland area | 30 m | - | Spatial |
| LU40 | % | Shrubland area | 30 m | - | Spatial |
| LU50 | % | Wetland area | 30 m | - | Spatial |
| LU60 | % | Waterbody area | 30 m | - | Spatial |
| LU80 | % | Artificial surface area | 30 m | - | Spatial |
| LU90 | % | Bareland area | 30 m | - | Spatial |
| LU100 | % | Permanent frozen land area | 30 m | - | Spatial |
| LU255 | % | Sea area | 30 m | - | Spatial |
| ROAD | Km grid$^{-1}$ | Road density | Polyline | - | Spatial |
| eBC | Mg grid$^{-1}$ | Emission of black carbon | 0.25° | Month | Spatial |
| eCO | Mg grid$^{-1}$ | Emission of CO | 0.25° | Month | Spatial |
| eCO2 | Mg grid$^{-1}$ | Emission of $CO_2$ | 0.25° | Month | Spatial |
| eNH3 | Mg grid$^{-1}$ | Emission of $NH_3$ | 0.25° | Month | Spatial |
| eNOx | Mg grid$^{-1}$ | Emission of $NO_2$ and NO | 0.25° | Month | Spatial |
| eOC | Mg grid$^{-1}$ | Emission of organic carbon | 0.25° | Month | Spatial |
| ePM25 | Mg grid$^{-1}$ | Emission of $PM_{2.5}$ | 0.25° | Month | Spatial |
| ePMcoar | Mg grid$^{-1}$ | Emission of PM-coarse | 0.25° | Month | Spatial |
| eSO2 | Mg grid$^{-1}$ | Emission of $SO_2$ | 0.25° | Month | Spatial |
| eVOC | Mg grid$^{-1}$ | Emission of VOC | 0.25° | Month | Spatial |

[a] Spatial or temporal resolution of raw data.

[b] Temporal: MOPITT is processed with the temporal and spatial convolution. Spatial: These variables have accompanying variables processed with the spatial convolution.

**Table S2.** Coverage rates of MOPITT-CO retrievals across China ($\mu \pm \sigma$; %)[a].

| Year(s) | Spring | Summer | Fall | Winter | Annual |
|---------|--------|--------|------|--------|--------|
| 2013 | 3.2 ± 2.1 | 3.0 ± 2.0 | 4.5 ± 2.6 | 4.1 ± 2.7 | 3.7 ± 0.6 |
| 2014 | 3.1 ± 2.1 | 2.7 ± 2.0 | 4.3 ± 2.5 | 4.1 ± 2.4 | 3.6 ± 0.6 |
| 2015 | 3.0 ± 2.0 | 3.0 ± 2.1 | 4.0 ± 2.4 | 3.8 ± 2.4 | 3.4 ± 0.6 |
| 2016 | 3.0 ± 2.1 | 3.0 ± 2.0 | 4.0 ± 2.7 | 3.6 ± 2.3 | 3.4 ± 0.6 |
| 2013-2016 | 3.1 ± 1.6 | 2.9 ± 1.5 | 4.2 ± 1.9 | 3.9 ± 2.0 | 3.5 ± 0.5 |

[a] $\sigma$ stands for the spatial variation. Please refer to Fig. S2 for the coverage maps.

**Table S3.** Correlations between the daily CO observations from the monitoring network and the MOPITT surface retrievals for China during 2013-2016.

| Region[a] | Spring | Summer | Fall | Winter | 2013 | 2014 | 2015 | 2016 | 2013-2016 |
|---|---|---|---|---|---|---|---|---|---|
| Central China | 0.08 | 0.18 | 0.21 | 0.27 | 0.59 | 0.22 | 0.32 | 0.24 | 0.30 |
| East China | 0.12 | 0.24 | 0.36 | 0.37 | 0.46 | 0.34 | 0.48 | 0.40 | 0.43 |
| North China | 0.30 | 0.23 | 0.31 | 0.42 | 0.36 | 0.31 | 0.39 | 0.38 | 0.37 |
| Northeast China | 0.28 | 0.34 | 0.38 | 0.30 | 0.28 | 0.41 | 0.39 | 0.38 | 0.39 |
| Northwest China | 0.31 | 0.12 | 0.27 | 0.31 | 0.52 | 0.39 | 0.35 | 0.31 | 0.38 |
| South China | 0.49 | 0.53 | 0.59 | 0.52 | 0.65 | 0.56 | 0.55 | 0.57 | 0.58 |
| Southwest China | 0.10 | 0.12 | 0.25 | 0.20 | 0.14 | -0.02 | 0.17 | 0.22 | 0.17 |
| Nation | 0.22 | 0.31 | 0.32 | 0.34 | 0.42 | 0.31 | 0.3 | 0.34 | 0.44 |
| Central Tibetan Plateau | 0.15 | 0.25 | 0.11 | 0.05 | -0.19 | -0.09 | -0.10 | 0.14 | -0.12 |
| North China Plain | 0.18 | 0.13 | 0.25 | 0.35 | 0.33 | 0.30 | 0.27 | 0.36 | 0.30 |

[a] Please refer to Fig. 1 for the locations of these regions.

**Table S4.** Comparisons of the RF and RF-STK models in predicting daily ground-level CO concentrations across China during 2013-2016 based on the 10-fold cross-validation.

| Metric[a] | $RF_r$[b] | $RF$[b] | $RF_{rw}$[b] | $RF_w$[b] | $RF_w$-STK[b] | $RF_{rw}$-STK[bc] | $RF_{rw}$-STK[b] |
|---|---|---|---|---|---|---|---|
| $R^2$ | 0.56 | 0.53 | 0.54 | 0.53 | 0.49 | 0.49 | 0.51 |
| Slope | 0.60 | 0.55 | 0.57 | 0.55 | 0.63 | 0.60 | 0.64 |
| RMSE | 0.50 | 0.52 | 0.51 | 0.52 | 0.55 | 0.57 | 0.54 |
| RPE | 46.1% | 48.0% | 47.1% | 48.1% | 51.0% | 53.3% | 50.4% |
| MFB | 0.0832 | -0.013 | -0.0076 | -0.0128 | -0.030 | 0.064 | -0.022 |
| MFE | 0.31 | 0.31 | 0.31 | 0.31 | 0.36 | 0.37 | 0.35 |
| MNB | 0.90 | 0.64 | 0.66 | 0.64 | 0.68 | 0.74 | 0.70 |
| MNE | 1.09 | 0.90 | 0.91 | 0.90 | 0.97 | 1.0 | 0.98 |

[a] $R^2$: coefficient of determination; RMSE: root mean square error (mg m$^{-3}$); RPE: relative prediction error; MFB: mean fractional bias; MFE: mean fractional error; MNB: mean normalized bias; MNE: mean normalized error.

[b] RF: random forest; STK: spatiotemporal kriging. Subscript r indicates a reduced model through variable selection, and subscript w means that the training samples were inversely weighted by the associated population densities. The CO concentrations were log-transformed to train all the models except for $RF_r$ which was trained with the CO concentrations at native scale.

[c] This RF-STK model was developed with the a priori information rather than the MOPITT retrievals.

**Table S5.** Previous studies modeling surface CO concentrations.

| Reference | Model[a] | Study Area | Study Period | Evaluation Metric[b] |
|---|---|---|---|---|
| (Hooghiemstra et al., 2012) | 4D-Var system | Bahia, Brazil | 2007-2009 | $R$=0.6 (daily; prior) $R$=0.8 (daily; posterior) |
| (Yeganeh et al., 2012) | SVM; PLS-SVM | Tehran, Iran | 2007.01-2011.01 | $R^2$=0.56 (daily; SVM) $R^2$=0.65 (daily; PLS-SVM) |
| (Hu et al., 2016) | CMAQ | China | 2013.03-2013.12 | MNE=0.59~0.66 (daily) MFE=0.86~1.02 (daily) |

[a] 4D-Var system: Four-dimensional variational data assimilation system; SVM: support vector machine; PLS-SVM: hybrid model of partial least square and support vector machine; CMAQ: Community Multiscale Air Quality model.

[b] All these studies conducted validation at daily level. Both prior and posterior estimates of the model were evaluated with an independent dataset (Hooghiemstra et al., 2012). Goodness of fit was evaluated in (Yeganeh et al., 2012), and an independent dataset was used for validation in (Hu et al., 2016).

[Figure]

**Figure S1:** Average diurnal pattern in CO concentrations across 1656 monitoring sites for China during 2013-2016. The peak and the valley appeared at 9am and 4pm (Beijing Standard Time). The shaded area represents the standard deviations.

[Figure]

**Figure S2:** Seasonal averages of the ground-level CO concentrations (mg m$^{-3}$) for the whole China, the North China Plain (NCP), the Central Tibetan Plateau (CTP), Naqu, and Qamdo during 2013-2016 based on (a) the MOPITT retrieved surface CO and (b) the observations from the monitoring network. The error bars represent the standard deviations. Naqu and Qamdo are two main cities in CTP (Fig. 8).

[Figure]

**Figure S3:** Coverage rates (%) of MOPITT-CO retrievals for (a) spring, (b) summer, (c) fall, and (d) winter during 2013-2016 across China. The coverage rate at each grid cell was calculated as the percentage of days with MOPITT-CO retrievals in each season.

[Figure]

**Figure S4:** Seasonal means of the averaging-kernel row-sum values associated with the MOPITT retrieved surface CO for (a) spring, (b) summer, (c) fall, and (d) winter during 2013-2016 across China. Small row-sum values indicate strong dependence of the MOPITT retrievals on the a priori information. Please refer to "S.2 Averaging kernel" for more explanation.

[Figure]

**Figure S5:** Annual averages of the MOPITT retrieved surface CO concentrations for (a) 2013, (b) 2014, (c) 2015, and (d) 2016.

[Figure]

**Figure S6:** Annual total CO emissions (t) in (a) 2013, (b) 2014, (c) 2015, and (d) 2016 from anthropogenic sources across China. Due to the data availability, the CO emissions for 2013 and 2015 were linearly interpolated from the available data for 2012, 2014, and 2016 (Li et al., 2017).

[Figure]

**Figure S7:** Evolution of the cross-validation RMSE (mg m$^{-3}$) and $R^2$ for the random forest submodels through the stepwise backward variable selection process.

[Figure]

**Figure S8:** Annual average ground-level CO concentrations from 2013 to 2016 predicted by the $RF_r$, RF, $RF_w$ and $RF_{rw}$-STK models with the MOPITT retrievals. RF: random forest; STK: spatiotemporal kriging. Subscript r indicates a reduced model through variable selection, and subscript w means that the training samples were inversely weighted by the associated population densities. The CO concentrations were log-transformed to train all the models except for $RF_r$ which was trained with the CO concentrations at native scale. The predictions for 2013-2016 by each model are presented from top to bottom rows.

[Figure]

**Figure S9:** Performance of the RF-STK model in predicting daily CO concentrations by regions, years, and seasons. The mean and standard deviation of the root mean square error (RMSE, mg m$^{-3}$) over all the 10-fold cross-validations are presented. The numbers of monitoring sites in Central, East, North, Northeast, Northwest, South, and Southwest China are 267, 255, 307, 171, 159, 278, and 219, respectively. The numbers of monitoring sites in 2013, 2014, 2015, and 2016 are 743, 1041, 1542, and 1603, respectively.

[Figure]

**Figure S10:** Partial dependence plots of the random forest submodel for delineating the relationship between each predictor variable and the ground-level CO concentrations. Partial dependence (Y axis) is the effect of a predictor variable (X axis) on the CO concentrations when the values of all the other predictor variables are fixed at their averages (Friedman, 2001). The subplots are arranged in the order of variable importance. Please refer to Table S1 for the descriptions and units of the predictor variables. The rug plot indicates the data density. Note that the partial dependence estimations are of high uncertainty given low data densities.

[Figure]

**Figure S11:** Coal consumption amounts and energy conversion rates in the sector of power generation and heating for China during 2013-2016 (CSY, 2018).

[Figure]

**Figure S12:** Correlations among the predictor variables and the ground-level CO concentrations, which were measured by the Spearman's rank correlation coefficients. Please refer to Table S1 for the detailed descriptions of the variables.

[Figure]

**Figure S13:** Anthropogenic emission sources of (a) CO, (b) organic carbon (OC), (c) black carbon (BC), (d) volatile organic compound (VOC), (e) NH$_3$, and (f) SO$_2$ for China during 2013-2016 (Li et al., 2017).

[Figure]

**Figure S14:** Total CO emissions (million t) from (a) industry, (b) power, (c) residential, and (d) transportation sectors in each season over China during 2013-2016 (Li et al., 2017).

[Figure]

**Figure S15:** Temporal trends of the population-weighted average ground-level CO concentrations (mg m$^{-3}$) for China during 2013-2016 based on the actual MOPITT retrieved surface CO (blue solid line), the MOPITT a priori surface CO (purple solid line), the predictions made by the RF-STK model using the actual MOPITT retrieved surface CO (red solid line), and the predictions made by the RF-STK model using the MOPITT a priori surface CO (black solid line). The points in different colors represent the deseasonalized monthly averages for deriving the corresponding trend lines. The 95% confidence intervals of the trends are in parentheses followed by the *P* values.

[Figure]

**Figure S16:** Temporal trends of the population-weighted average ground-level CO concentrations (mg m$^{-3}$) for China during 2013-2016 based on the actual (blue solid line) and the bias-adjusted (green solid line) MOPITT retrieved surface CO, as well as the predictions made by the RF-STK model using the actual MOPITT retrieved surface CO (red solid line). The bias correction was carried out according to the mean bias drift of -0.69% per year reported in the previous study (Deeter et al., 2017). The points in different colors represent the deseasonalized monthly averages for deriving the corresponding trend lines. The 95% confidence intervals of the trends (mg m$^{-3}$ per year) are in parentheses followed by the *P* values.

---

## Author Response (AR2)

**Response to Co-Editor's comments:**

Although one reviewer recommended rejecting your manuscript, that reviewer did not identify any concerns that could not be addressed but just noted that "Revisions would be likely to involve considerable additional effort ". Based on your comprehensive responses to all of the reviewer comments/concerns, the manuscript has been substantially improved. I have just a few minor comments regarding your responses (see below) that I would like you to address before the manuscript can be accepted for publication in ACP.

**Response:** We greatly appreciate your consideration and comments on our manuscript. We have addressed all of the comments in the revised manuscript. Please see the point-by-point response (in blue) below.

1) Page 9, Lines 37-38; Page 10, Lines 1-6.

"It is noteworthy that the RF-STK model with MOPITT-CO was superior to the model without MOPITT-CO ($R^2$=0.49, RMSE=0.58 mg m$^{-3}$, and slope=0.60) and the model with the a priori information ($R^2$=0.49, RMSE=0.57 mg m$^{-3}$, and slope=0.60) based on the site-based cross-validation results (Tables 2 and S4). The performance difference became more apparent in the region-based cross-validation, where the model with MOPITT-CO ($R^2$=0.45, RMSE=0.61 mg m$^{-3}$, and slope=0.52) clearly outperformed the model without MOPITT ($R^2$=0.32, RMSE=0.69 mg m$^{-3}$, and slope=0.46). We therefore reasoned that the MOPITT-CO data were essential for the RF-STK model to achieve better predictive performance, especially for the areas without monitoring sites nearby."

Note that the numbers in that first sentence appear to be incorrect (they differ from what is in Table 2) but even with the correct numbers I don't see how it is justified to say that the model with MOPITT-CO is "superior" unless you can show this is a significant difference. The region-based difference does show more improvement and the description of this in the second and third sentences is justified.

**Response:** We apologize for the unclear expression, but the numbers in that first sentence are correct. The numbers in the first pair of parentheses ("$R^2$=0.49, RMSE=0.58 mg m$^{-3}$, and slope=0.60" from the third column of Table 2) are for the RF-STK model without MOPITT-CO. The numbers in the second pair of parentheses ("$R^2$=0.49, RMSE=0.57 mg m$^{-3}$, and slope=0.60" from the seventh column of Table S4) are for the RF-STK model with the a priori information. As the performance results for the RF-STK model with MOPITT-CO have been mentioned before this sentence (Page xx, Lines xx-xx; "$R^2$=0.51, RMSE=0.54 mg m$^{-3}$, and slope=0.64" from the second column of Table 2), we did not restate the numbers in this sentence, which caused this confusion. In the revision, we have reworded this sentence for clarity. Please see Page 9, Lines 37-38; and Page 10, Lines 1-2.

"It is noteworthy that the RF-STK model with MOPITT-CO ($R^2$=0.51, RMSE=0.54 mg m$^{-3}$, and slope=0.64; Table 2) was superior to the model without MOPITT-CO ($R^2$=0.49, RMSE=0.58 mg m$^{-3}$, and slope=0.60; Table 2) and the model with the a priori information ($R^2$=0.49, RMSE=0.57 mg m$^{-3}$, and slope=0.60; Table S4) based on the site-based cross-validation results."

2) Page 11, Lines 37-38; Page 12, Lines 1-3.

"The trend underestimation by MOPITT-CO might be largely due to the setting that the a priori information was the same across the years"
This sentence is confusing and should be reworded.
**Response:** Suggestion is taken. We have revised these sentences for clarity. Please see Page 11, Lines 37-38; and Page 12, Lines 1-4.

"We found that the trend underestimation tended to be more severe for the regions with weaker averaging kernels, indicating higher dependence on the a priori information (Figs. 9 and S5). In addition, the decreasing trend predicted by the RF-STK model with the a priori information (-2.06% per year) was slower than that predicted by the RF-STK model with the MOPITT-CO (-2.25% per year; Fig. S15). We thus deduced that the a priori information, which was the same across the years (Dekker et al., 2017), might greatly contribute to the trend underestimation by the MOPITT-CO."

3) Page 12, Line 14-18.
"We chose the period of 2013-2016 for this study due to the data availability".
While I realize that you don't have data to go earlier than 2013, why not include 2017?
**Response:** We started to conduct this study in the beginning of 2018. The 2017 data for all the variables of the RF-STK model were not completely available until the summer of 2018. As it would take a long time to rerun the whole analyses, we did not include the 2017 data in this study. Thank you for your understanding. We have revised this sentence to address this limitation. Please see Page 12, Lines 15-16.

"We chose the period of 2013-2016 due to the data availability in the beginning of 2018 when we started to conduct this study."

4) Page 12, Line 24.
"Since July of 2018, the TROPOspheric Monitoring Instrument onboard the Sentinel-5P satellite has been providing the CO product at a higher resolution of 7 km × 3.5 km (Borsdorff et al., 2018), which may replace MOPITT-CO in the RF-STK model in order to make predictions at a higher resolution."
Revise the last part of this sentence to: "which could replace MOPITT-CO in the RF-STK model in order to make predictions for years after 2018 at a higher resolution."
**Response:** Suggestion is taken. In the revision, we have revised the last part of this sentence. Please see Page 12, Lines 26-27.

[revised manuscript text omitted]